# BabyVision: Visual Reasoning Beyond Language

Liang Chen [1 2 *]  Weichu Xie [1 2 *]  Yiyan Liang [1 2 *]  Hongfeng He [1 *]  Haozhe Zhao [1 *]  Zhibo Yang [3]
Zhiqi Huang [4]  Haoning Wu [4]  Haoyu Lu [4]  Y. charles [4]  Yiping Bao [4]  Yuantao Fan [5]  Guopeng Li [5]
Haiyang Shen [1 2]  Xuanzhong Chen [1 6]  Wendong Xu [1]  Shuzheng Si [6]  Zefan Cai [7]  Wenhao Chai [8]  Ziqi Huang [9]
Fangfu Liu [6]  Tianyu Liu [2]  Baobao Chang [2]  Ming Wu [10]  Xiaobo Hu [11]  Kaiyuan Chen [11]  Yixin Ren [11]
Yang Liu [11]  Yuan Gong [11]  Kuan Li [1]

## Abstract

While humans develop core visual skills long before acquiring language, contemporary Multimodal LLMs (MLLMs) still rely heavily on linguistic priors to compensate for their fragile visual understanding. We uncovered a crucial fact: state-of-the-art MLLMs consistently fail on basic visual tasks that humans, even 3-year-olds, can solve effortlessly. To systematically investigate this gap, we introduce BABYVISION, a benchmark designed to assess core visual abilities independent of linguistic knowledge for MLLMs. BABYVISION spans a wide range of tasks, with 388 items divided into 22 subclasses across four key categories. Empirical results and human evaluation reveal that leading MLLMs perform significantly below human baselines. Gemini3-Pro-Preview scores 49.7, falling well behind the average adult score of 94.1. These results show despite excelling in knowledge-heavy evaluations, current MLLMs still lack fundamental visual primitives. Progress in BABYVISION represents a step toward human-level visual perception and reasoning capabilities. We also explore solving visual reasoning with generation models by proposing BABYVISION-GEN and automatic evaluation toolkit. Code and data are released at BabyVision.

## 1. Introduction

Human visual understanding emerges remarkably early in life—well before children acquire language or formal symbolic reasoning. Within their first few months, infants can already discriminate between shapes and textures, track moving objects, infer occlusion and depth, and form expectations about simple physical events (Johnson, 2011; Braddick & Atkinson, 2011; Kellman et al., 2006).

These *early-vision abilities* form a foundation for later cognitive functions—including language, abstract reasoning, and motor planning—by supporting structured representations of the physical world. In contrast, contemporary Multimodal Large Language Models (MLLMs) (Team et al., 2023; OpenAI, 2025; Bai et al., 2025) appear to exhibit an inverted profile of competence. They achieve strong performance on many high-level, knowledge-intensive benchmarks that typically demand substantial education or domain expertise (e.g., HLE (Team, 2025b), MMMU (Yue et al., 2024)), including multi-step geometric and mathematical reasoning (e.g., MathVista (Lu et al., 2024), MathVision (Wang et al., 2024)), large-scale recognition of people and places (e.g., MME (Fu et al., 2024a)), and even medically styled question answering (e.g., DrVD-Bench (Zhou et al., 2025)). Yet, despite these successes, our studies reveal a consistent weakness in *basic* visual tasks that children (ages 3–12) can solve with little or no language mediation—such as visual discrimination, visual tracking, spatial understanding, and simple visual pattern recognition. Even the strongest model we tested, Gemini3-Pro-Preview, remains approximately 20% behind 6-year-old children in our pilot study, as shown in Figure 1. This discrepancy suggests that current MLLMs still lack the atomic visual competencies that underlie human vision from its earliest stages. However, existing evaluations rarely target these **beyond-language visual understanding** abilities in a systematic way.

To fill this gap, we introduce BABYVISION, a benchmark designed with a scientific and rigorous data curation pipeline to probe the atomic visual skills humans develop early in life. BABYVISION aims to minimize reliance on linguistic knowledge by emphasizing tasks driven by perception and pattern regularities rather than textual reasoning or semantic priors. The benchmark contains 388 unique questions spanning 22 subclasses across four domains: *Fine-grained*

---

*Equal contribution  [1]UniPat AI [2]Peking University [3]Alibaba Group [4]MoonShot AI [5]StepFun [6]Tsinghua University [7]University of Wisconsin–Madison [8]Princeton University [9]Nanyang Technological University [10]0G Labs [11]xbench. Correspondence to: Liang Chen <liangchen@unipat.ai>, Kuan Li <kuanli@unipat.ai>.

*Proceedings of the 43rd International Conference on Machine Learning*, Seoul, South Korea. PMLR 306, 2026. Copyright 2026 by the author(s).

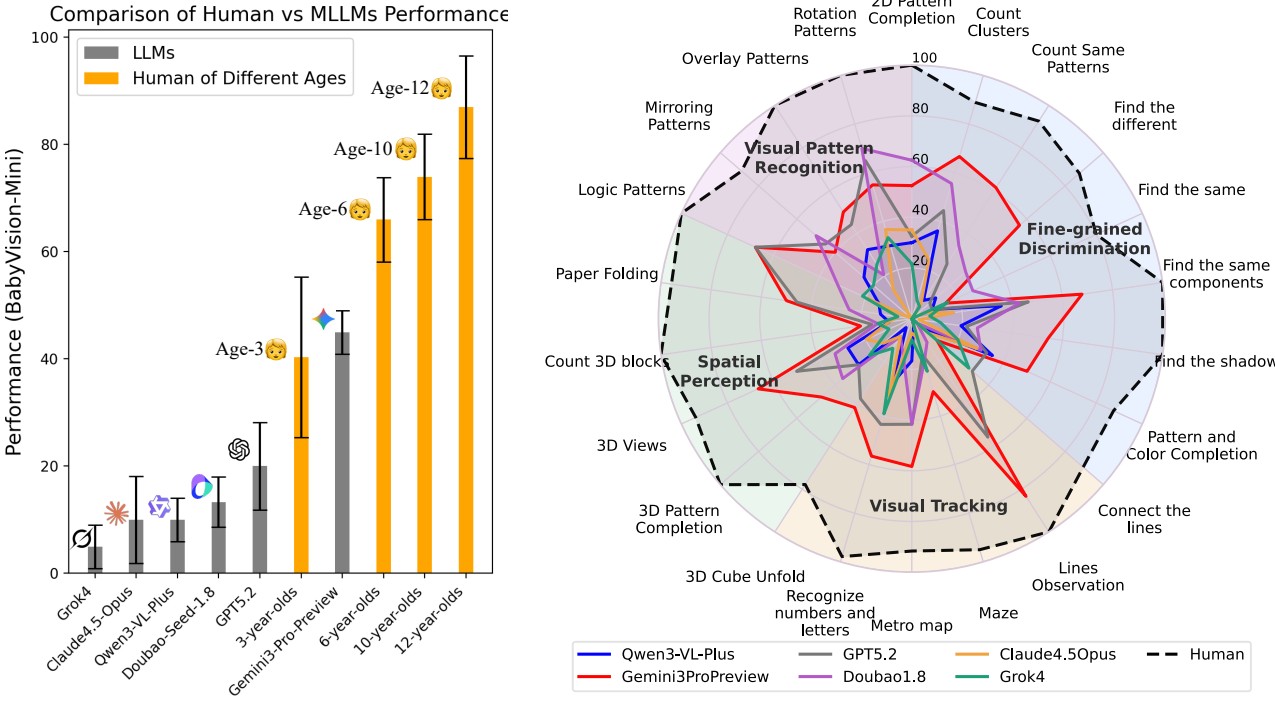

*Figure 1.* Performance on BABYVISION among MLLMs and human of different ages.

*Discrimination*, *Visual Tracking*, *Spatial Perception*, and *Visual Pattern Recognition*. Together, these domains cover a broad spectrum of early-vision competencies with substantial diversity in visual conditions and task structure.

As shown in Figure 1, we evaluate state-of-the-art MLLMs on BABYVISION, covering both proprietary and open models (e.g., Gemini3-Pro-Preview, GPT-5.2, Qwen3VL-235B-Thinking). In general, we observe a substantial gap between current MLLMs and human performance on early-vision competencies. Specifically, the best-performing model achieves an overall accuracy of **49.7%**, whereas adult human testers reach **94.1%** (a **44.4%** absolute gap), with consistent deficits across all four domains. The largest failures appear in *Visual Tracking* and *Spatial Perception*, where models frequently exhibit errors such as a *loss of manifold identity* (e.g., losing track of curves through intersections) and a *failure of spatial imagination* (e.g., the inability to mentally transform 3D structures). These results indicate that strong high-level multimodal reasoning does not imply robust foundational visual competence, and that BABYVISION exposes a distinct failure mode not captured by existing benchmarks.

Beyond textual evaluation, we observe that many questions from BABYVISION are most naturally solved visually: humans typically draw on image—by tracing paths, completing patterns, or marking spatial relations—rather than verbalizing their reasoning. Therefore, we introduce BABYVISION-GEN, a generative counterpart to BABYVI-

SION that evaluates visual reasoning through visual generation **beyond language output**, and develop an automatic evaluation tool achieving a 96% agreement rate with human judgments. Experiments with frontier image and video generators (e.g., Nano-Banana-Pro, Sora-2) show promising gains on tasks that are challenging for MLLMs, especially visual tracking and fine-grained discrimination, though overall reliability and solution consistency remain limited.

In summary, this paper makes three main contributions: (1) BABYVISION, a benchmark with 388 questions across 22 subclasses in four domains for evaluating beyond-language visual reasoning; (2) BABYVISION-GEN, an extension for evaluating visual reasoning ability for generation models **beyond language output**, with an automatic evaluation toolkit; and (3) comprehensive evaluation revealing a performance landscape marked by a substantial 44.4% human-model gap and a systematic investigation of model failure modes, with fine-grained failure analysis and investigation of RLVR training for visual reasoning.

## 2. Related Work

### 2.1. Benchmarks for MLLMs

The rapid progress of Multimodal Large Language Models (MLLMs) has driven the development of diverse evaluation benchmarks. A prominent line of work targets expert-level, knowledge-intensive multimodal reasoning. MMMU (Yue

**Input Image**

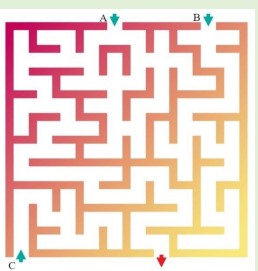

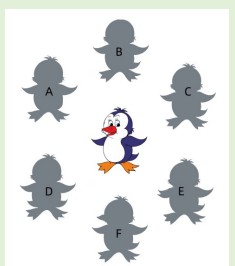

**BabyVision: Question and Answer**

*Q:* The figure shows forty-nine tiger patterns arranged in 7 rows and 7 columns, among which one is different from the others. In which row and column is it located? Answer in the format (row,column).
*A:* (4,7)

*Q:* Find the entrance connected to the exit among the three entrances. Choose from A, B and C.
*A:* Entrance A

*Q:* Three animal patterns (A, B, C) and three environment patterns (1, 2, 3) are connected by lines. How are they paired by lines? Answer in the format: A-1, B-2, C-3
*A:* A-2,B-3,C-1

*Q:* Find the shadow that perfectly matches the penguin image in the middle.
*A:* Shadow C

**BabyVision-Gen: Question and Answer**

*Q:* Put a red circle on the unique element in the picture.
*A:*



*Q:* The entrance at A is the right one to the exit. Please draw in red lines showing the way out of the maze from the entrance to the exit.
*A:*

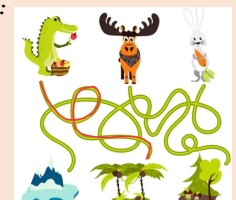

*Q:* Draw a red line to trace the complete line extending from the top left figure.
*A:*

*Q:* Put a red circle on the shadow that exactly matches the penguin.
*A:*

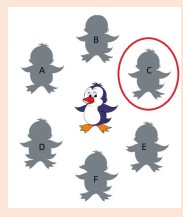

*Figure 2.* Examples of BABYVISION and BABYVISION-GEN. While BABYVISION evaluates visual understanding through language output, BABYVISION-GEN evaluates visual reasoning through image generation.

et al., 2024) assembles 11.5K college-level questions spanning 30 disciplines, while HLE (Team, 2025b) pushes further toward "humanity's last exam" with expert-level problems. Mathematical reasoning benchmarks such as MathVision (Wang et al., 2024), and MathVerse (Zhang et al., 2024) focus on geometric and quantitative reasoning grounded in images, while MME (Fu et al., 2024a) and MMEval-Pro (Huang et al., 2025) provide comprehensive evaluations across perception and cognition. ReasonMap (Feng et al., 2026) utilizes transit maps to evaluate fine-grained visual and spatial reasoning in MLLMs, while subsequent work (Feng et al., 2025) employs multi-stage reinforcement learning to enhance these capabilities. Leading models achieve impressive results—Gemini 3 Pro attains 92.8% on MMMU—demonstrating the increasingly effective integration of visual inputs with domain knowledge.

However, strong results on these benchmarks do not necessarily translate into robust visual perception. This inverted competence profile—where models excel at expert-level tasks but struggle with basic perception—has been documented across several studies. BLINK (Fu et al., 2024b) reveals a substantial gap between humans (95.7%) and leading

MLLMs on classical vision problems that require no specialized knowledge. MMStar (Chen et al., 2024) demonstrates that models can achieve 42.9% on MMMU *without* visual input. MMVP (Tong et al., 2024) further probes visual limitations and finds that models fail to distinguish images with clear perceptual differences. SpatialViz-Bench (Wang et al., 2025a) also finds top MLLMs far below a strong human baseline. These findings reveal a blind spot: existing benchmarks predominantly draw from tasks designed for experts and rely on semantic recognition rather than perceptual primitives. BABYVISION addresses this gap by targeting pre-linguistic visual abilities—foundational competencies humans develop before language acquisition—and comparing model performance directly to children aged 3–12.

**2.2. Early Vision and Developmental Foundations**

The design of BABYVISION draws on developmental psychology research establishing that humans acquire core visual competencies well before language. Infants demonstrate object permanence by 3–4 months (Baillargeon et al., 1985), track objects through occlusion (Johnson, 2011), and discriminate depth and shape before acquiring lan-

guage (Kellman et al., 2006; Braddick & Atkinson, 2011). The core knowledge hypothesis (Spelke, 2000) posits innate systems for representing objects, space, numbers, and agents—capacities that are independent of linguistic mediation. By ages 3–6, children perform complex visual discrimination, understand spatial transformations, and recognize patterns, often demonstrating competence through nonverbal responses (Karmiloff-Smith, 1994; Golomb, 2003). Developmental research also shows that children externalize visual reasoning through drawing before verbalizing solutions (Kellogg, 1969). This observation motivates These developmental trajectories motivates key design choices in both BABYVISION and BABYVISION-GEN.

## 3. BABYVISION

In this section, we present BABYVISION, a benchmark designed to evaluate early-vision capabilities of Multimodal Large Language Models (MLLMs). Unlike previous benchmarks that focus on complex semantic reasoning or domain knowledge, BABYVISION targets foundational visual skills that humans acquire during early development—abilities that emerge before or alongside language acquisition. We split vision-centric reasoning into four categories: *Fine-grained Discrimination* (detecting subtle visual differences; 8 subtypes), *Visual Tracking* (following paths, lines, and trajectories; 5 subtypes), *Spatial Perception* (understanding 3D structures and relationships; 5 subtypes), and *Visual Pattern Recognition* (identifying logical and geometric patterns; 4 subtypes). Together, these comprise 22 basic subtypes, each targeting a fundamental visual capability. Through a rigorous data curation pipeline, we construct 388 questions spanning a wide diversity of visual reasoning tasks.

### 3.1. Data Curation Pipeline

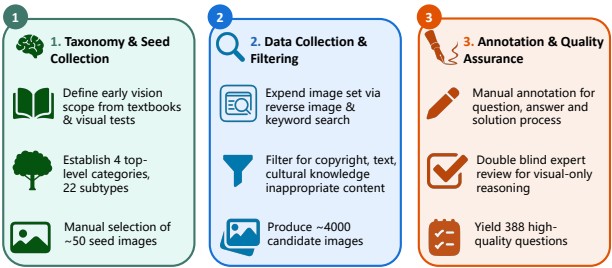

*Figure 3.* Overview of the multi-stage data collection and curation pipeline for BABYVISION: taxonomy design & seed selection, data augmentation & filtering, and annotation & quality assurance.

Our rigorous multi-stage data collection curation process (Figure 3) follows three steps to ensure quality and validity:

**Taxonomy Definition and Seed Collection.** We first define the scope of "early vision" grounded in developmental psychology. Consulting textbooks designed for children

under 12 and standardized visual development tests, we identify core visual competencies and establish four top-level categories with 22 specific subtypes. For each subtype, we manually select 4–5 "seed images" that exemplify the visual task, yielding approximately 100 high-quality seed examples that serve as prototypes for data expansion.

**Data Collection and Filtering.** Using the seed images, we expand the dataset through reverse image search and keyword-based retrieval, crawling similar images from the internet. We strictly adhere to copyright regulations, retaining only images permitted for academic use. Images containing substantial text, requiring cultural knowledge, or depicting inappropriate content are filtered out. This stage produces approximately 4,000 candidate images.

**Annotation and Quality Assurance.** Each candidate image undergoes manual annotation by trained annotators. Annotators first determine whether the image can support a meaningful visual question aligned with our taxonomy—images that rely heavily on text reading or require cultural background knowledge are discarded. For valid images, annotators write specific questions and answers and, crucially, provide a detailed "solution" proving the correctness.

All annotations then pass through a double-blind review: two independent experts verify that the answer is unambiguous and derivable largely from visual analysis rather than language. A question is included only if both experts agree on the answer and reasoning logic. Disagreements are returned to the original annotator for revision; questions that remain contentious after revision are permanently discarded.

### 3.2. Dataset Statistics

BABYVISION comprises 388 carefully curated questions, each paired with a unique image. The dataset includes 135 multiple-choice questions (34.8%) and 253 fill-in-the-blank questions (65.2%), with balanced answer distributions to mitigate position bias. The questions are concise (averaging 25.9 words) and tightly grounded in visual content, minimizing opportunities for language-based shortcuts. The distribution of task categories is shown in Figure 4. In addition, we collect 1,400 training examples following the same construction pipeline but sourced more broadly, enabling an investigation of how training influences model performance on BABYVISION, which is discussed in Section 6.2.

### 3.3. Evaluation

This encourages reasoning prior to the final response and facilitates automated answer extraction. By default, we select the highest reasoning effort of the tested models if adjustable. We evaluate model responses using an LLM-as-judge approach. Given a model's output answer and ground-truth answer, we query Qwen3-Max to determine semantic

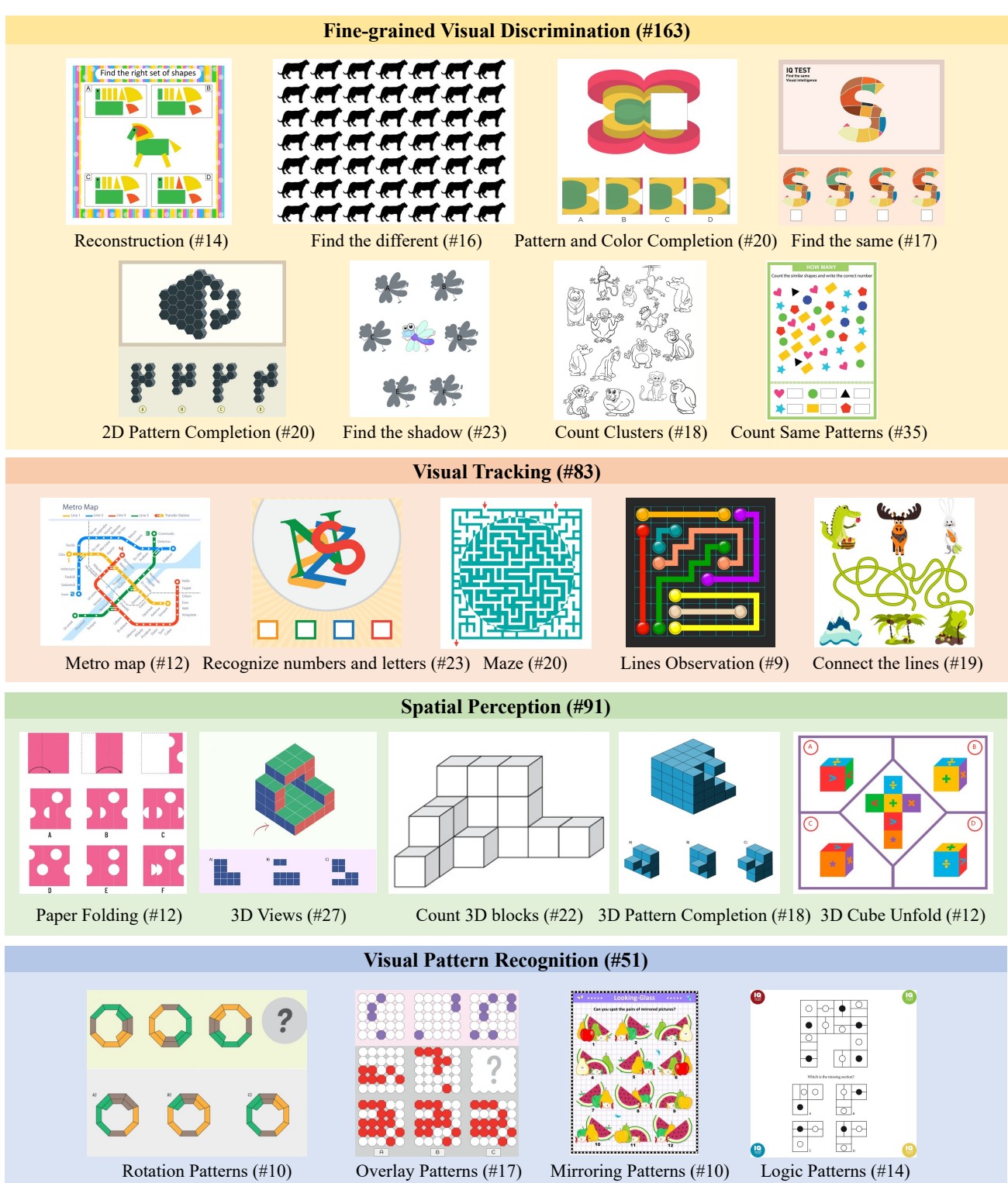

*Figure 4.* Example questions and the number of examples (#) from BABYVISION spanning the four core categories and 22 types.

equivalence, which shows 100% consistency with human evaluators. The judge prompt is provided in Appendix A.

# 4. BABYVISION-GEN

A key observation from BABYVISION is that many tasks admit solutions that are most naturally expressed *visually*. When humans solve maze navigation or pattern completion problems, they often draw trajectories or complete shapes rather than verbalizing step-by-step reasoning (Kellogg, 1969; Golomb, 2003). Similarly, in find-the-difference tasks, solutions are typically conveyed by directly circling the differing regions in the image. This observation motivates BABYVISION-GEN, a generative extension that evaluates whether models can perform visual reasoning through image (or video) generation, providing an alternative pathway that bypasses the verbalization bottleneck.

## 4.1. Visual Reasoning via Generation

BABYVISION-GEN adapts a subset of BABYVISION questions for visual output and alters the prompt to support generation. For each question, we provide a generation prompt instructing the model to produce an image that demonstrates the solution—tracing a path, completing a pattern, or marking spatial relationships. This formulation tests whether visual generation models can leverage visual reasoning capabilities that may exceed their ability to verbalize answers. We show some examples of BABYVISION-GEN in Figure 2.

**Statistics.** BABYVISION-GEN comprises 280 questions across 21 subtypes organized into the same four categories as BABYVISION: Fine-grained Discrimination (128 questions, 8 subtypes), Visual Tracking (55 questions, 4 subtypes), Spatial Perception (59 questions, 5 subtypes), and Visual Pattern Recognition (38 questions, 4 subtypes). Note that one subtype from BABYVISION is excluded as it does not naturally admit a generative solution format. Each instance includes the original image, a generation prompt (averaging 22.9 words), and a reference solution image for evaluation, reflecting the consensus of three human annotators. The detailed distribution across categories and subtypes is provided in Table 4 in Appendix B.4.

## 4.2. Evaluation

Evaluating BABYVISION-GEN requires determining whether a model's generated image expresses correct visual solutions. Therefore, we ask models to provide explicit visual answers by annotating the original input image with minimal overlays (e.g., circles, lines, arrows, or text). For automatic evaluation, we employ Gemini-3-Flash as a judge. The judge is given three images: (i) the input image, (ii) a human-annotated ground-truth solution, and (iii) the model-generated output. It returns a binary decision indicating whether the generated annotation matches the ground truth under subtype-specific criteria (e.g., identical selected option; same traced route for mazes). Full prompts

and criteria are provided in Appendix B.

To further assess the reliability of automatic evaluation, we perform human evaluation on all Nano-Banana-Pro outputs using PhD-level annotators. Automatic and human judgments agree on **96.1%** of instances (269/280; F1=0.924), supporting the use of the automatic judge for scalable evaluation. Subtype-wise agreement rates and the confusion matrix are reported in Appendix B.

# 5. Main Experiments

## 5.1. Evaluation Setup

**Models.** Our evaluation encompasses 13 frontier models, comprising 6 proprietary and 7 open-source systems. The proprietary models include Gemini3-Pro-Preview (Team et al., 2023), GPT-5.2 (OpenAI, 2025), Doubao-1.8 (Seed, 2025), Qwen3-VL-Plus (Bai et al., 2025), Claude-4.5-Opus (Anthropic, 2025), and Grok-4 (xAI, 2025). The open-source selection consists of Kimi-K2.5 (Moonshot AI, 2026), Qwen3VL-235B-Thinking (Bai et al., 2025), InternVL3.5-241B (Wang et al., 2025b), Step3 (StepFun et al., 2025), KimiVL-A3B (Team et al., 2025b), MimoVL-7B-RL (Team et al., 2025a), and GLM4.6V (Team, 2025a).

Furthermore, to examine the influence of model scaling and reasoning modes (thinking vs. non-thinking), we evaluate additional variants of the Qwen3VL family, including Qwen3VL-235B (Thinking/Instruct) and the smaller Qwen3VL-32B/8B/4B-Thinking models.

**Evaluation Settings.** We report the Avg@3 results for all evaluated models. For proprietary models, we utilize the official APIs without modifying default parameters, with the exception of reasoning effort, which is set to the maximum level where adjustable. For open-source models, we adhere to the inference temperatures recommended in their official papers; unless otherwise specified, we employ a default sampling temperature of 1.0. The inference and evaluation prompts for BABYVISION and BABYVISION-GEN are detailed in Section 3.3 and Section 4.2, respectively.

**Human Testers.** In our primary experiment, we conducted a comparative study using 20 representative samples (BabyVision-Mini) spanning diverse categories. We recruited participants across four age groups—3, 6, 10, and 12 years old—with 20 individuals per group[1]. The study was conducted with full permission from school authorities and parents, as well as the voluntary assent of the participants. Each group was allotted one class period (45 minutes) to complete the test. We conduct the evaluation of BABYVISION with 16 college-educated adults completed the full 388-question benchmark. Human testers are not included in

---

[1]Testers aged 3-12 are from a single school population. Results may vary across different populations or backgrounds.

| Type | Sub-Type | Human | Gemini3-Pro-Preview | | GPT-5.2 | | Doubao-1.8 | | Qwen3-VL-Plus | | Claude-4.5-Opus | | Grok-4 | |
|---|---|---|---|---|---|---|---|---|---|---|---|---|---|---|
| | | | Avg ($\mu$)↑ | Std ($\sigma$)↓ | Avg ($\mu$)↑ | Std ($\sigma$)↓ | Avg ($\mu$)↑ | Std ($\sigma$)↓ | Avg ($\mu$)↑ | Std ($\sigma$)↓ | Avg ($\mu$)↑ | Std ($\sigma$)↓ | Avg ($\mu$)↑ | Std ($\sigma$)↓ |
| Fine-grained Discrimination | 2D Pattern Completion | 100 | 52.5 | 2.5 | 32.5 | 7.5 | **62.5** | 2.5 | 30.0 | 5.0 | 35.0 | 4.1 | 21.7 | 4.7 |
| | Count Clusters | 88.9 | **66.7** | 0.0 | 44.4 | 5.6 | 55.6 | 5.6 | 36.1 | 2.8 | 24.1 | 2.6 | 7.4 | 2.6 |
| | Count Same Patterns | 92.7 | **61.4** | 4.3 | 25.7 | 2.9 | 34.3 | 5.7 | 8.6 | 2.9 | 6.7 | 1.4 | 5.7 | 0.0 |
| | Find the Different | 87.5 | **56.3** | 6.3 | 9.4 | 3.1 | 28.1 | 3.1 | 12.5 | 0.0 | 0.0 | 0.0 | 0.0 | 0.0 |
| | Find the Same | 80 | 14.7 | 2.9 | 8.8 | 2.9 | **26.5** | 2.9 | 8.8 | 8.8 | 5.9 | 0.0 | 15.7 | 2.8 |
| | Reconstruction | 100 | **67.9** | 3.6 | 46.4 | 3.6 | 42.9 | 0.0 | 35.7 | 0.0 | 16.7 | 3.4 | 7.1 | 5.8 |
| | Find the Shadow | 100 | **54.4** | 6.5 | 21.7 | 0.0 | 26.1 | 4.4 | 19.6 | 10.9 | 2.9 | 2.1 | 11.6 | 7.4 |
| | Pattern and Color Completion | 87.5 | **50.0** | 10.0 | 32.5 | 7.5 | 30.0 | 5.0 | 35.0 | 0.0 | 28.3 | 2.4 | 20.0 | 4.1 |
| | **Sub-Type Overall Results** | 92.3 | **46.2** | 5.5 | 27.3 | 0.9 | 39.2 | 0.0 | 21.8 | 2.2 | 14.3 | 0.3 | 11.0 | 2.7 |
| Visual Tracking | Connect the Lines | 89.5 | 13.2 | 2.6 | **31.6** | 0.0 | 2.6 | 2.6 | 5.3 | 5.3 | 8.8 | 6.6 | 29.8 | 2.5 |
| | Lines Observation | 100 | **83.3** | 5.6 | 55.6 | 0.0 | 11.1 | 11.1 | 0.0 | 0.0 | 0.0 | 0.0 | 0.0 | 0.0 |
| | Maze | 95.0 | **30.0** | 5.0 | 15.0 | 0.0 | 17.5 | 2.5 | 2.5 | 2.5 | 8.3 | 2.4 | 21.7 | 4.7 |
| | Metro Map | 91.7 | **58.3** | 8.3 | 41.7 | 0.0 | 41.7 | 8.3 | 16.7 | 0.0 | 6.3 | 3.9 | 8.3 | 6.8 |
| | Recognize Numbers and Letters | 97.8 | **56.5** | 4.4 | 43.5 | 0.0 | 13.0 | 0.0 | 26.1 | 4.4 | 31.9 | 2.1 | 39.1 | 0.0 |
| | **Sub-Type Overall Results** | 94.6 | **43.4** | 2.4 | 34.9 | 0.0 | 15.7 | 1.2 | 11.5 | 0.6 | 13.7 | 2.8 | 24.1 | 1.0 |
| Spatial Perception | 3D Cube Unfold | 77.8 | **41.7** | 8.3 | 37.5 | 4.2 | 8.3 | 0.0 | 4.2 | 4.2 | 8.3 | 6.8 | 13.9 | 10.4 |
| | 3D Pattern Completion | 100.0 | **47.2** | 13.9 | 27.8 | 0.0 | 36.1 | 8.3 | 27.8 | 5.6 | 14.8 | 2.6 | 22.2 | 7.9 |
| | 3D Views | 93.8 | **66.7** | 7.4 | 50.0 | 1.9 | 33.3 | 3.7 | 27.8 | 5.6 | 19.8 | 6.3 | 9.9 | 4.6 |
| | Count 3D Blocks | 100 | **20.5** | 2.3 | 15.9 | 6.8 | 13.6 | 4.6 | 9.1 | 4.6 | 9.1 | 3.7 | 13.6 | 3.7 |
| | Paper Folding | 95.8 | **50.0** | 0.0 | 45.8 | 12.5 | 25.0 | 0.0 | 12.5 | 4.2 | 5.6 | 3.9 | 5.6 | 3.9 |
| | **Sub-Type Overall Results** | 94.7 | **53.7** | 1.5 | 35.2 | 2.2 | 24.7 | 0.6 | 18.1 | 1.7 | 12.8 | 1.4 | 13.2 | 3.2 |
| Visual Pattern Recognition | Logic Patterns | 100 | **67.9** | 3.6 | **67.9** | 3.6 | 32.1 | 3.6 | 14.3 | 7.1 | 19.1 | 3.4 | 21.4 | 5.8 |
| | Mirroring Patterns | 88.9 | 40.0 | 10.0 | 45.0 | 5.0 | **50.0** | 0.0 | 25.0 | 5.0 | 0.0 | 0.0 | 20.0 | 0.0 |
| | Overlay Patterns | 100 | **50.0** | 2.9 | 44.1 | 2.9 | 20.6 | 2.9 | 32.4 | 14.7 | 13.7 | 5.6 | 25.5 | 2.8 |
| | Rotation Patterns | 100 | 55.0 | 15.0 | 65.0 | 15.0 | **70.0** | 0.0 | 30.0 | 10.0 | 36.7 | 12.5 | 33.3 | 9.4 |
| | **Sub-Type Overall Results** | 97.8 | 53.9 | 1.0 | **54.9** | 5.9 | 37.7 | 1.5 | 25.5 | 3.9 | 17.0 | 3.7 | 24.8 | 2.5 |
| All | **Overall Results** | 94.1 | **49.7** | 2.6 | 34.4 | 1.7 | 30.2 | 0.3 | 19.2 | 0.6 | 14.2 | 1.2 | 16.2 | 1.3 |

*Table 1.* **Performance (Avg@3) of Proprietary MLLMs on BabyVision.** The best results for each question type are marked in **bold**. Reported values represent the average Pass@1 accuracy across three random runs, accompanied by the standard deviation. All models are in thinking mode with highest reasoning budget.

the evaluation of BABYVISION-GEN.

### 5.2. BABYVISION Results

**Proprietary Model Performance.** The strongest proprietary model performance remains substantially below the human baseline (94.1%). Among proprietary systems, Gemini3-Pro-Preview achieves the highest overall score (49.7%), followed by GPT-5.2 (34.4%) and Doubao-1.8 (30.2%), while remaining models trail by a wide margin (e.g., Qwen3-VL-Plus at 19.2%, Grok-4 at 16.2%, Claude-4.5-Opus at 14.2%). The large gap to human performance is consistent across all task families, with no single category dominating the error profile. It suggests that current systems still lack foundational visual competencies, pointing to a systemic limitation rather than an isolated weakness.

At the subtype level, several tasks prove challenging for nearly all systems. In particular, *Count 3D Blocks* yields uniformly low accuracy (best: 20.5%), indicating that compositional 3D counting under occlusion and viewpoint variation remains difficult. Similarly, *Find the Same* reaches only 26.5% at best (Doubao-1.8), despite a human baseline of 80%. Such failures are informative because they reflect deficiencies in structured scene representations (e.g., object permanence and depth-aware composition), rather than shortcomings in recognition or superficial pattern matching.

**Open-source Models and Scaling.** We also evaluate open-source models and investigate model scaling effects in Tables 6 and 7 in the Appendix. Among open-source models, Kimi-K2.5 achieves the best performance at 32.1%, substantially outperforming other open-source systems (e.g.,

Qwen3VL-235B-Thinking at 22.2%). While Kimi-K2.5 narrows the gap with proprietary models, it still falls short of Gemini3-Pro-Preview by 17.6 points. For other open-source models, test-time thinking provides consistent gains, but scaling alone does not guarantee improvement—the 4B-Thinking model slightly outperforms 8B-Thinking.

**Comparison with Young Humans.** As shown in Figure 1, most frontier MLLMs perform well below the average 3-year-old on BabyVision-Mini, despite their PhD-level results on language benchmarks. Gemini3-Pro-Preview is the only model consistently above the Age-3 band, yet it still lags typical 6-year-olds by about 20 points. This highlights a core limitation: the issue is not solving "hard problems," but struggling with pre-language visual primitives—the early perceptual and spatial abilities humans acquire before language becomes the main reasoning tool.

**Fine-grained Analysis.** Subtype-level analysis reveals systematic failure patterns across all four domains. *Visual Tracking* exposes fundamental weaknesses: most models score near zero on *Lines Observation*, frequently "switching tracks" at intersections—errors immediately obvious to humans. *Spatial Perception* is similarly challenging, with *Count 3D Blocks* yielding only 20.5% accuracy even for the best model, as tasks requiring 3D mental models resist verbalization. *Fine-grained Discrimination* shows variable performance, with models struggling most on *Find the Same* (best: 26.5%) where subtle visual differences must be detected. In contrast, *Visual Pattern Recognition* shows relatively stronger performance when patterns involve discrete, rule-based transformations like rotation. Detailed

| Type | Sub-Type | # | NanoBanana-Pro | | GPT-Image-1.5 | | Qwen-Image-Edit-2511 | |
|------|----------|---|---------------|---|--------------|---|---------------------|---|
| | | | Avg ($\mu$) ↑ | Std ($\sigma$) ↓ | Avg ($\mu$) ↑ | Std ($\sigma$) ↓ | Avg ($\mu$) ↑ | Std ($\sigma$) ↓ |
| Fine-grained Discrimination | 2D Pattern Completion | 19 | **24.6** | 4.8 | 12.3 | 2.5 | 14.0 | 6.6 |
| | Count Clusters | 9 | **22.2** | 9.1 | 3.7 | 5.2 | 7.4 | 5.2 |
| | Count Same Patterns | 31 | **31.2** | 6.1 | 0.0 | 0.0 | 2.2 | 3.0 |
| | Find the Different | 16 | **35.4** | 2.9 | 12.5 | 8.8 | 2.1 | 2.9 |
| | Find the Same | 10 | 3.3 | 4.7 | **3.3** | 4.7 | 0.0 | 0.0 |
| | Reconstruction | 13 | **30.8** | 6.3 | 10.3 | 3.6 | 0.0 | 0.0 |
| | Find the Shadow | 14 | **9.5** | 3.4 | 7.1 | 5.8 | 2.4 | 3.4 |
| | Pattern and Color Completion | 16 | 22.9 | 2.9 | **31.3** | 0.0 | 8.3 | 5.9 |
| | **Category Average** | 128 | **24.5** | 1.8 | 9.6 | 1.2 | 4.7 | 0.8 |
| Visual Tracking | Connect the Lines | 12 | 0.0 | 0.0 | 0.0 | 0.0 | 0.0 | 0.0 |
| | Maze | 18 | 0.0 | 0.0 | 0.0 | 0.0 | 0.0 | 0.0 |
| | Metro Map | 12 | **11.1** | 3.9 | 5.6 | 3.9 | 0.0 | 0.0 |
| | Recognize Numbers and Letters | 13 | **17.9** | 3.6 | 5.1 | 3.6 | 0.0 | 0.0 |
| | **Category Average** | 55 | **6.7** | 2.1 | 2.4 | 1.0 | 0.0 | 0.0 |
| Spatial Perception | 3D Cube Unfold | 12 | 2.8 | 3.9 | 0.0 | 0.0 | 0.0 | 0.0 |
| | 3D Pattern Completion | 18 | 5.6 | 0.0 | **14.8** | 2.6 | 13.0 | 5.2 |
| | 3D Views | 19 | **21.1** | 8.6 | 17.5 | 2.5 | 10.5 | 4.5 |
| | Count 3D Blocks | 5 | **26.7** | 9.4 | 20.0 | 0.0 | 0.0 | 0.0 |
| | Paper Folding | 5 | **20.0** | 16.3 | 6.7 | 9.4 | 0.0 | 0.0 |
| | **Category Average** | 59 | **13.0** | 6.0 | 12.4 | 2.6 | 7.3 | 3.5 |
| Visual Pattern Recognition | Logic Patterns | 11 | **30.3** | 8.6 | 12.1 | 8.6 | 12.1 | 11.2 |
| | Mirroring Patterns | 3 | 0.0 | 0.0 | **44.4** | 15.7 | 11.1 | 15.7 |
| | Overlay Patterns | 15 | **24.4** | 16.5 | 8.9 | 2.9 | 0.0 | 0.0 |
| | Rotation Patterns | 9 | 18.5 | 5.2 | **25.9** | 5.2 | 14.8 | 10.5 |
| | **Category Average** | 38 | **22.8** | 10.0 | 16.7 | 5.5 | 7.9 | 2.6 |
| All | **Overall** | 280 | **18.3** | 2.2 | 9.8 | 1.0 | 4.8 | 0.7 |

*Table 2.* **Performance (%) of Visual Generation Models on BABYVISION-GEN.** The best results for each subtype are marked in **bold**. # denotes the number of questions in each subtype. Reported values represent the average accuracy across three random runs, accompanied by the standard deviation.

per-category analysis is provided in Appendix C.3.

### 5.3. BABYVISION-GEN Results

We evaluate three frontier visual generation models on BABYVISION-GEN: NanoBanana-Pro, GPT-Image-1.5, and Qwen-Image-Edit. Results are presented in Table 2. Note that since BABYVISION-GEN uses a different evaluation methodology (visual output assessment via image generation) and covers a subset of tasks adapted for generative evaluation, the results are not directly comparable to the MLLM experiments on BABYVISION.

**Overall Performance.** NanoBanana-Pro achieves highest overall accuracy at 18.3%, substantially outperforming GPT-Image-1.5 (9.8% ) and Qwen-Image-Edit (4.8%). The relatively low standard deviations across runs suggest consistent behavior, though overall accuracy remains limited.

**Subtype Analysis.** Several subtypes reveal the characteristics of visual generation approaches. On *Find the Different*, NanoBanana-Pro achieves 35.4%, and *Count Same Patterns* reaches 31.2%. These tasks share a common characteristic: solutions are naturally expressed by drawing or marking on the image. For *Spatial Perception* tasks such as *Count 3D Blocks* (26.7%), NanoBanana-Pro demonstrates non-trivial performance. In *Visual Pattern Recognition*, the models show mixed results: *Logic Patterns* (30.3%) and *Overlay Patterns* (24.4%) yield moderate accuracy, while *Mirror-*

*ing Patterns* proves challenging (0% for NanoBanana-Pro, though GPT-Image-1.5 achieves 44.4% with high variance). These results suggest that visual generation offers a promising but currently limited pathway for visual reasoning tasks.

## 6. Discussion

### 6.1. Why Do Frontier Models Fail on Simple Tasks?

Our evaluation reveals a striking inversion: state-of-the-art MLLMs that excel at PhD-level reasoning struggle with visual tasks that young children solve effortlessly. Qualitative analysis identifies four systematic failure modes, illustrated in Figure 7: (1) *loss of fine-grained detail*, where MLLMs fail to distinguish candidates based on sub-semantic cues (e.g., curvature or pixel-level alignment) due to the degradation of fine-grained visual information; (2) *loss of manifold identity*, where topological consistency breaks down, rendering models unable to track continuous manifolds (e.g., winding lines) through occlusions; (3) *failure of spatial imagination*, where MLLMs struggle to perform 3D mental transformations (e.g., rotation, projection) as language proves an insufficient coordinate system for stable 3D modeling; and (4) *failure of visual pattern induction*, where models fail to acquire abstract rules from sparse examples, often conflating surface attributes with underlying transformation logic. Details are provided in Appendix D.

A unifying theme is what we term the *verbalization bot-*

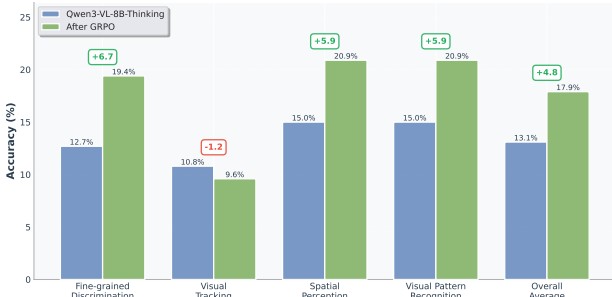

*Figure 5.* Performance of Qwen3-VL-8B-Thinking before vs. after RLVR fine-tuning on BabyVision. RLVR yields a +4.8 overall accuracy gain and improves most subtypes, except Visual Tracking.

*tleneck*: MLLMs process visual inputs by translating them into language representations before reasoning. While leveraging the powerful reasoning capabilities of LLMs, it introduces a fundamental limitation: visual information that cannot be faithfully expressed in language. Consider the distinction between *describable* and *undescribable* visual properties. Semantic content like "a red car" translates well to language, but geometric relationships (precise curvature, intersection positions) resist verbalization. BabyVision targets these *undescribable* properties, explaining why language-heavy benchmark performance does not transfer. We validate this causally: disabling CoT on Qwen3-VL-8B-Thinking yields +1.3% overall accuracy, and attention probing shows vision token attention increases 5.28× without CoT (Appendix C.5), confirming that linguistic reasoning diverts the model from visual features. This suggests progress on early-vision tasks may require architectures that preserve visual information throughout reasoning, rather than compressing it into a linguistic bottleneck.

### 6.2. Can RLVR Help Visual Reasoning?

To bridge performance gaps observed in BabyVision, we investigate whether Reinforcement Learning with Verifiable Rewards(RLVR)—leveraging its success in language reasoning (Xie et al., 2026)—can enhance visual capabilities. Specifically, we fine-tune Qwen3-VL-8B-Thinking via GRPO on 1,400 training examples generated through a similar pipeline.

As shown in Figure 5, RLVR yields a **+4.8 point** overall improvement on BabyVision, with consistent gains across most tasks. However, Visual Tracking shows minimal improvement, and in some subtypes even degrades. This aligns with our verbalization bottleneck hypothesis: RLVR enhances performance by encouraging longer, more structured reasoning in language, which benefits tasks amenable to verbal decomposition but provides limited help for continuous perceptual tracking that resists verbalization.

This finding suggests that while RLVR offers a promising direction for improving visual reasoning, addressing the fundamental perceptual limitations exposed by BabyVision may require approaches beyond language-mediated reasoning. Full details and analysis are provided in Appendix C.4.

### 6.3. Beyond Language: Visual Externalization

A key observation motivating BABYVISION-GEN is that many BabyVision tasks are most naturally solved by *visual externalization*—drawing, marking, or tracing on the image rather than verbalizing. This suggests an alternative paradigm: models could reason in visual space and output visual solutions. Our experiments with generation models on BABYVISION-GEN reveal promising signals (Figure 8 in Appendix E): models like Sora-2 and NanoBanana-Pro exhibit human-like visual thinking, explicitly drawing trajectories along paths, though their solutions still contain errors indicating that generation must be guided by robust visual understanding.

This points to *visual generation models as multimodal reasoners*. Recent unified architectures like Bagel (ByteDance Seed, 2025) maintain visual representations throughout reasoning, enabling explicit thinking in visual space. Similarly, frontier video generators like Sora 2 (OpenAI, 2025) and Veo 3 (Google DeepMind, 2025) demonstrate emergent capabilities in modeling spatial relationships, allowing generation itself to serve as visual reasoning (Cai et al., 2025). If models can learn to manipulate images directly—tracing paths, completing patterns, transforming shapes—they may bypass the verbalization bottleneck entirely

## 7. Conclusion

We present BabyVision, a benchmark targeting the early-vision abilities that humans develop before language acquisition—capabilities that current MLLMs surprisingly lack. Our evaluation exposes a 44.4% accuracy gap between the state-of-the-art model and human, with MLLMs often performing below 3-year-old children on tasks requiring fine-grained discrimination, visual tracking, spatial perception, and pattern recognition. These failures trace to a common root: the *verbalization bottleneck*, where visual information that cannot be faithfully expressed in language is discarded during reasoning. This finding carries an important implication: excelling on language-heavy benchmarks does not guarantee robust visual foundations. We further introduce BABYVISION-GEN to evaluate visual reasoning through generation, revealing that models capable of "thinking in pixels" show promise on tasks where verbal reasoning fails. Together, BabyVision and BABYVISION-GEN provide diagnostic tools for tracking progress toward multimodal systems with genuinely grounded visual intelligence.

## Impact Statement

BabyVision exposes a fundamental gap in current MLLMs: despite impressive performance on language-heavy and expert-level benchmarks, these models lack robust foundational visual competence—often falling below the level of young children on pre-linguistic visual primitives. By decomposing visual intelligence into atomic capabilities benchmarked independently of language, our work reveals where current models fail and demonstrates that scaling language-mediated reasoning alone is insufficient. Our findings further suggest that visual generation—reasoning by drawing, tracing, and manipulating images—offers a promising alternative that can partially recover capabilities beyond the reach of text-based reasoning. These foundational visual abilities are also prerequisites for embodied AI: reliable physical-world assistance demands visual competence exceeding that of a three-year-old. BabyVision thus provides both a diagnostic lens for current limitations and a research direction for future models—to advance multimodal intelligence, we must rebuild vision from the ground up rather than relying on linguistic shortcuts.

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

## A. Evaluation Details for BABYVISION

We employ Qwen3-Max as the judge model to evaluate semantic equivalence between model outputs and ground-truth answers. The judge receives the following prompt:

---

EVALUATION PROMPT FOR BABYVISION

```
You are a careful and strict evaluator. You will be given:
1. QUESTION
2. GROUND TRUTH ANSWER (correct answer)
3. MODEL OUTPUT (answer from another model)
YOUR GOAL: Determine if the Model Output accurately matches the Ground Truth Answer in meaning.
MATCHING MEANS: the facts, entities, and key details are equivalent, even if phrasing differs.
NOT MATCHING MEANS: the Model Output is wrong, incomplete, contains extra incorrect facts, or changes
the meaning.
PROCESS (INTERNAL REASONING):
  1. Read and understand the Question, Ground Truth Answer, and Model Output.
  2. Ignore small wording differences, formatting, or synonyms.
  3. If all factual content matches, conclude True. Otherwise, conclude False.
IMPORTANT: Think through your decision step-by-step internally before responding. In your final output,
return only True or False, with no extra text or explanation.
OUTPUT FORMAT: True or False
INPUT:
Question: {Question}
Ground Truth Answer: {Groundtruth}
Model Output: {Model Output}
```

---

## B. Evaluation Details for BABYVISION-GEN

### B.1. Inference Protocol

For BABYVISION-GEN, we instruct visual generation models to annotate the original image with their solution. This ensures that the original visual context is preserved while allowing models to demonstrate their reasoning through visual marks. We use the following prompt template:

---

**Generation Prompt Template**

```
CRITICAL INSTRUCTION: You are a visual annotation assistant. Your task is to add ONLY the requested
annotations (circles, lines, arrows, text labels) to mark the answer on the image.
IMPORTANT RULES:
  1. DO NOT modify, redraw, or alter ANY part of the original image content
  2. DO NOT change colors, shapes, positions, or any visual elements of the original image
  3. ONLY add overlay annotations (circles, lines, arrows, text) on TOP of the original image
  4. The original image must remain 100% intact and unchanged
  5. Use bright, visible colors (red, green, blue) for your annotations so they stand out
  6. Keep annotations minimal and precise - only mark what is asked
YOUR TASK: {Question}
REMINDER: Keep the original image EXACTLY as it is. ONLY add annotation marks (circles, lines, arrows,
or text) to indicate your answer. Do not redraw or modify any part of the original image.
```

---

### B.2. Automatic Evaluation Protocol

We employ Gemini-3-Flash as the judge model to evaluate whether generated images correctly solve the visual reasoning task. The evaluation is conducted by comparing three images: (1) the original input image, (2) the ground-truth solution image annotated by human experts, and (3) the model-generated image. The judge determines whether the generated solution matches the ground truth.

We design type-specific evaluation criteria for each subtype to ensure accurate assessment. The general prompt structure is:

**Automatic Evaluation Prompt**

You are evaluating an AI-generated image for a visual reasoning task.
**TASK TYPE:** {task_type}
**SUBTYPE:** {subtype}
**GENERATION INSTRUCTION:** "{generation_prompt}"
**You are provided with THREE images:**
- Image 1 (Input): The original question/puzzle image
- Image 2 (Ground Truth): The CORRECT answer showing what the result SHOULD look like
- Image 3 (Generated): The AI-generated result to be evaluated

Compare Image 3 (Generated) with Image 2 (Ground Truth) to determine if they show the SAME answer.
**[Type-Specific Criteria Inserted Here]**
**DECISION RULES:**
- TRUE: Generated image shows the EXACT SAME answer as Ground Truth
- FALSE: Generated shows a DIFFERENT answer, NO answer, or UNCLEAR answer
**IMPORTANT:**
- Focus ONLY on whether the ANSWER matches, ignore style differences
- A marking on a DIFFERENT element/option = FALSE
- A path taking a DIFFERENT route = FALSE
- A DIFFERENT number or character = FALSE
- Missing required answer = FALSE
**Respond with ONLY one word:** "True" or "False"

## B.3. Type-Specific Evaluation Criteria

We customize evaluation criteria for each task subtype. The complete criteria are provided below:

### B.3.1. FINE-GRAINED DISCRIMINATION

**Find the Different**

**TASK:** Find the unique/different element among many similar elements.
**CRITERIA:** Is the circle on the SAME grid position as ground truth?
- Circle on DIFFERENT element = FALSE
- No circle visible = FALSE

**Find the Same**

**TASK:** Find identical elements or matching figures.
**CRITERIA:** Are ALL marked elements the same as in ground truth?
- Circles on DIFFERENT elements = FALSE
- Missing circles = FALSE

**Find the Shadow**

**TASK:** Find the shadow/silhouette that matches the colored figure.
**CRITERIA:** Is the SAME option circled?

**Reconstruction / 2D Pattern Completion / Pattern and Color Completion**

**TASK:** Find/select the correct option.
**CRITERIA:** Is the SAME option circled/selected?

**Count Same Patterns / Count Clusters**

**TASK:** Count patterns or fill in numbers.
**CRITERIA:** Do the markings/numbers match ground truth exactly?

### B.3.2. VISUAL TRACKING

**Maze**

**TASK:** Draw a path through the maze.
**CRITERIA:** Does the path follow the EXACT SAME route as ground truth?
- Different route = FALSE
- No visible path = FALSE

**Connect the Lines**

**TASK:** Trace a line following the continuous path.
**CRITERIA:** Does the traced line follow the SAME path as ground truth?

**Metro Map**

**TASK:** Draw the shortest path between metro stations.
**CRITERIA:** Does the path follow the EXACT SAME route as ground truth?

**Recognize Numbers and Letters**

**TASK:** Fill in letters/numbers in blanks.
**CRITERIA:** Are the EXACT SAME characters filled in each blank?

### B.3.3. SPATIAL PERCEPTION

**3D Views / 3D Cube Unfold / Paper Folding / 3D Pattern Completion**

**TASK:** Select the correct option for spatial reasoning.
**CRITERIA:** Is the SAME option circled?

**Count 3D Blocks**

**TASK:** Count cubes in a 3D structure.
**CRITERIA:** Is the EXACT SAME number written?

### B.3.4. VISUAL PATTERN RECOGNITION

**Logic Patterns / Mirroring Patterns / Overlay Patterns / Rotation Patterns**

**TASK:** Identify pattern and select correct option.
**CRITERIA:** Is the SAME option circled?

### B.4. BABYVISION-GEN Statistics

Table 4 presents the detailed distribution of questions across categories and subtypes in BABYVISION-GEN.

### B.5. Human Evaluation and Validation

To validate the reliability of our automatic evaluation pipeline, we conducted comprehensive human evaluation on all 280 NanoBanana-Pro outputs. PhD-level annotators independently judged each generated image against the ground truth.

**Overall Agreement.** The automatic scorer achieves **96.1%** agreement with human judgments (269/280 samples). The confusion matrix is shown in Table 5.

**Evaluation Metrics.** The key metrics are summarized below:

| Type | Sub-Type | Kimi-K2.5 | | Qwen3VL-235B-Thinking | | InternVL3.5-241B | | GLM4.6V | | MimoVL-7B-RL | | Step3 | | KimiVL-A3B | |
|---|---|---|---|---|---|---|---|---|---|---|---|---|---|---|---|
| | | Avg ($\mu$) ↑ | Std ($\sigma$) ↓ | Avg ($\mu$) ↑ | Std ($\sigma$) ↓ | Avg ($\mu$) ↑ | Std ($\sigma$) ↓ | Avg ($\mu$) ↑ | Std ($\sigma$) ↓ | Avg ($\mu$) ↑ | Std ($\sigma$) ↓ | Avg ($\mu$) ↑ | Std ($\sigma$) ↓ | Avg ($\mu$) ↑ | Std ($\sigma$) ↓ |
| Fine-grained Discrimination | 2D Pattern Completion | **45.0** | 10.8 | 25.0 | 10.8 | 11.7 | 6.2 | 40.0 | 10.8 | 30.0 | 10.8 | 33.3 | 9.4 | 25.0 | 8.2 |
| | Count Clusters | **42.6** | 2.6 | 27.8 | 0.0 | 20.4 | 9.4 | 20.4 | 2.6 | 11.1 | 4.5 | 9.3 | 6.9 | 3.7 | 2.6 |
| | Count Same Patterns | **25.7** | 7.0 | 15.2 | 3.6 | 10.5 | 2.7 | 6.7 | 2.7 | 9.5 | 1.4 | 3.8 | 2.7 | 1.9 | 1.4 |
| | Find the Different | **29.2** | 7.8 | 16.7 | 3.0 | 20.8 | 3.0 | 14.6 | 3.0 | 2.1 | 3.0 | 0.0 | 0.0 | 6.3 | 0.0 |
| | Find the Same | 21.6 | 5.6 | **23.5** | 8.3 | 9.8 | 5.6 | 9.8 | 2.8 | 7.8 | 2.8 | 2.0 | 2.8 | 3.9 | 5.6 |
| | Reconstruction | **50.0** | 15.4 | 33.3 | 3.4 | 38.1 | 3.4 | 26.2 | 3.4 | 9.5 | 3.4 | 11.9 | 8.9 | 14.3 | 5.8 |
| | Find the Shadow | 14.5 | 7.4 | 15.9 | 5.4 | **17.4** | 3.6 | 7.3 | 2.1 | 7.3 | 2.1 | 10.1 | 2.1 | 7.3 | 5.4 |
| | Pattern and Color Completion | **46.7** | 6.2 | 38.3 | 9.4 | 31.7 | 6.2 | 25.0 | 7.1 | 35.0 | 7.1 | 21.7 | 6.2 | 13.3 | 8.5 |
| | **Sub-Type Overall Results** | **32.9** | 1.0 | 23.3 | 0.9 | 18.6 | 2.3 | 17.4 | 2.9 | 14.1 | 2.7 | 11.3 | 2.3 | 8.8 | 1.5 |
| Visual Tracking | Connect the Lines | 0.0 | 0.0 | 5.3 | 4.3 | 15.8 | 4.3 | 14.0 | 2.5 | 3.5 | 2.5 | 7.0 | 2.5 | **24.6** | 2.5 |
| | Lines Observation | **44.4** | 0.0 | 7.4 | 5.2 | 0.0 | 0.0 | 0.0 | 0.0 | 0.0 | 0.0 | 0.0 | 0.0 | 0.0 | 0.0 |
| | Maze | **26.7** | 2.4 | 11.7 | 6.2 | 23.3 | 2.4 | 21.7 | 2.4 | 8.3 | 2.4 | 21.7 | 2.4 | 23.3 | 4.7 |
| | Metro Map | **27.8** | 7.9 | 13.9 | 3.9 | 2.8 | 3.9 | 0.0 | 0.0 | 13.9 | 10.4 | 2.8 | 3.9 | 5.6 | 3.9 |
| | Recognize Numbers and Letters | 23.2 | 7.4 | 36.2 | 7.4 | 39.1 | 3.6 | 30.4 | 6.2 | 14.5 | 2.1 | **53.6** | 4.1 | 8.7 | 3.6 |
| | **Sub-Type Overall Results** | **21.7** | 3.6 | 16.9 | 4.3 | 20.5 | 2.0 | 16.9 | 1.7 | 8.8 | 2.1 | 22.1 | 1.5 | 14.5 | 2.0 |
| Spatial Perception | 3D Cube Unfold | **33.3** | 6.8 | 19.4 | 7.9 | 8.3 | 6.8 | 8.3 | 6.8 | 8.3 | 6.8 | 11.1 | 10.4 | 2.8 | 3.9 |
| | 3D Pattern Completion | **42.6** | 11.4 | 29.6 | 6.9 | 22.2 | 4.5 | 33.3 | 7.9 | 37.0 | 2.6 | 25.9 | 9.4 | 31.5 | 14.6 |
| | 3D Views | **34.6** | 3.5 | 29.6 | 8.0 | 18.5 | 3.0 | 22.2 | 3.0 | 23.5 | 1.8 | 9.9 | 4.6 | 19.8 | 4.6 |
| | Count 3D Blocks | **19.7** | 5.7 | 10.6 | 2.1 | 6.1 | 2.1 | 6.1 | 4.3 | 10.6 | 4.3 | 3.0 | 2.1 | 6.1 | 4.3 |
| | Paper Folding | 13.9 | 3.9 | 8.3 | 6.8 | 11.1 | 3.9 | 16.7 | 0.0 | 13.9 | 3.9 | 11.1 | 7.9 | **19.4** | 3.9 |
| | **Sub-Type Overall Results** | **29.7** | 0.9 | 20.9 | 2.7 | 13.9 | 2.9 | 18.0 | 1.4 | 19.8 | 0.9 | 11.7 | 1.4 | 16.5 | 2.4 |
| Visual Pattern Recognition | Logic Patterns | **40.5** | 6.7 | 19.1 | 6.7 | 26.2 | 3.4 | 11.9 | 3.4 | 9.5 | 3.4 | 14.3 | 0.0 | 4.8 | 3.4 |
| | Mirroring Patterns | **56.7** | 4.7 | 26.7 | 9.4 | 23.3 | 4.7 | 23.3 | 4.7 | 26.7 | 4.7 | 6.7 | 4.7 | 6.7 | 9.4 |
| | Overlay Patterns | **49.0** | 11.1 | 33.3 | 7.3 | 25.5 | 2.8 | 13.7 | 7.3 | 21.6 | 2.8 | 31.4 | 2.8 | 21.6 | 14.0 |
| | Rotation Patterns | **63.3** | 9.4 | 40.0 | 0.0 | 43.3 | 12.5 | 33.3 | 4.7 | 26.7 | 12.5 | 16.7 | 4.7 | 16.7 | 9.4 |
| | **Sub-Type Overall Results** | **51.0** | 7.3 | 29.4 | 5.8 | 28.8 | 1.9 | 19.0 | 2.5 | 20.3 | 3.3 | 19.0 | 0.9 | 13.1 | 4.9 |
| All | **Overall Results** | **32.1** | 1.6 | 22.2 | 1.0 | 19.2 | 0.7 | 17.6 | 1.8 | 15.1 | 1.3 | 14.7 | 0.8 | 12.4 | 1.7 |

*Table 3.* **Performance (Avg@3) of Open-Source MLLMs on BabyVision.** The best results for each question type are marked in **bold**. Reported values represent the average Pass@1 accuracy across three random runs, accompanied by the standard deviation.

- **Human Evaluation Accuracy:** 75/280 (26.8%)
- **Auto Evaluation Accuracy:** 70/280 (25.0%)
- **Precision** (auto correct when human correct): 0.957
- **Recall** (finds human correct cases): 0.893
- **F1 Score:** 0.924

# C. Additional Experimental Results

## C.1. Open-source Model Performance

Table 6 presents the detailed performance of open-source MLLMs on BabyVision. Kimi-K2.5 achieves the best open-source performance at 32.1%, substantially outperforming other open-source systems such as Qwen3VL-235B-Thinking (22.2%) and InternVL3.5-241B (19.2%). Notably, Kimi-K2.5 excels particularly on Visual Pattern Recognition (51.0%), approaching proprietary model performance in this category. However, even the best open-source model remains 17.6 points behind Gemini3-Pro-Preview (49.7%), indicating that the capability gap persists. For the Qwen3VL family, test-time thinking consistently provides measurable gains (e.g., 22.2% vs. 19.5% for Thinking vs. Instruct at 235B).

## C.2. Impact of Model Scaling

Table 7 shows how model size and reasoning strategy affect performance within the Qwen3VL family. The Thinking variant consistently outperforms the Instruct variant at the same scale (e.g., 22.2% vs. 19.5% at 235B). Across model sizes, performance generally improves with scale: 235B-Thinking (22.2%) outperforms 32B-Thinking (17.4%), which exceeds 8B-Thinking (13.1%). However, the 4B-Thinking model (14.6%) slightly exceeds 8B-Thinking, suggesting that scaling alone does not guarantee monotonic improvement on fundamental perceptual tasks.

## C.3. Fine-grained Analysis by Category

We analyze performance at the subtype level across all four domains to identify which visual primitives are most challenging for current MLLMs.

**Fine-grained Discrimination.** Within this category, models show highly variable performance across subtypes. Gemini3-Pro-Preview achieves relatively strong results on *Reconstruction* (67.9%) and *Count Clusters* (66.7%), but all models struggle with *Find the Same* (best: 26.5%) and *Find the Different* (best: 56.3%). These tasks require detecting subtle visual differences among highly similar patterns—a capability that demands high-fidelity perceptual encoding rather than semantic abstraction.

| Category | Subtype | # |
|---|---|---|
| Fine-grained Discrimination | 2D Pattern Completion | 19 |
| | Count Clusters | 9 |
| | Count Same Patterns | 31 |
| | Find the Different | 16 |
| | Find the Same | 10 |
| | Reconstruction | 13 |
| | Find the Shadow | 14 |
| | Pattern and Color Completion | 16 |
| | *Subtotal* | *128* |
| Visual Tracking | Connect the Lines | 12 |
| | Maze | 18 |
| | Metro Map | 12 |
| | Recognize Numbers and Letters | 13 |
| | *Subtotal* | *55* |
| Spatial Perception | 3D Cube Unfold | 12 |
| | 3D Pattern Completion | 18 |
| | 3D Views | 19 |
| | Count 3D Blocks | 5 |
| | Paper Folding | 5 |
| | *Subtotal* | *59* |
| Visual Pattern Recognition | Logic Patterns | 11 |
| | Mirroring Patterns | 3 |
| | Overlay Patterns | 15 |
| | Rotation Patterns | 9 |
| | *Subtotal* | *38* |
| **Total** | | **280** |

*Table 4.* Distribution of questions across categories and subtypes in BABYVISION-GEN. One subtype from BABYVISION is excluded as it does not naturally admit a generative solution format.

**Visual Tracking.** This category exposes a fundamental weakness in maintaining continuous identity across spatial trajectories. The *Lines Observation* task is particularly diagnostic: Gemini3-Pro-Preview achieves 83.3%, but most other models score near zero, indicating a complete inability to trace curves through intersections. Similarly, *Connect the Lines* and *Maze* tasks reveal that models frequently "switch tracks" at crossings—errors that are immediately obvious to human observers.

**Spatial Perception.** The *Count 3D Blocks* subtype is uniformly challenging across all models, with the best accuracy at only 20.5% (Gemini3-Pro-Preview). This task requires inferring hidden volume and maintaining a coherent 3D mental model—capabilities that cannot be verbalized without information loss. In contrast, *3D Views* (66.7% for Gemini3-Pro-Preview) proves more tractable, likely because the viewpoint transformation can be partially reasoned through geometric rules.

**Visual Pattern Recognition.** Models show comparatively stronger performance in this category, with both Gemini3-Pro-Preview and GPT-5.2 achieving 67.9% on *Logic Patterns* and Doubao-1.8 reaching 70.0% on *Rotation Patterns*. These results suggest that when patterns involve discrete, rule-based transformations (rotation, reflection), models can partially leverage their symbolic reasoning capabilities. However, *Overlay Patterns* (best: 50.0%) and *Mirroring Patterns* (best: 50.0%) remain challenging compared to the near-perfect human baselines (100% and 88.9% respectively), indicating that continuous spatial transformations are still problematic.

|  | Human: Correct | Human: Incorrect | Total |
|---|---|---|---|
| **Auto: Correct** | 67 | 3 | 70 |
| **Auto: Incorrect** | 8 | 202 | 210 |
| **Total** | 75 | 205 | 280 |

*Table 5.* Confusion matrix for automatic vs. human evaluation on NanoBanana-Pro outputs. Green cells indicate agreement; red cells indicate disagreement.

| Type | Sub-Type | Kimi-K2.5 | | Qwen3VL-235B-Thinking | | InternVL3.5-241B | | GLM4.6V | | MimoVL-7B-RL | | Step3 | | KimiVL-A3B | |
|---|---|---|---|---|---|---|---|---|---|---|---|---|---|---|---|
| | | Avg ($\mu$) ↑ | Std ($\sigma$) ↓ | Avg ($\mu$) ↑ | Std ($\sigma$) ↓ | Avg ($\mu$) ↑ | Std ($\sigma$) ↓ | Avg ($\mu$) ↑ | Std ($\sigma$) ↓ | Avg ($\mu$) ↑ | Std ($\sigma$) ↓ | Avg ($\mu$) ↑ | Std ($\sigma$) ↓ | Avg ($\mu$) ↑ | Std ($\sigma$) ↓ |
| Fine-grained Discrimination | 2D Pattern Completion | **45.0** | 10.8 | 25.0 | 10.8 | 11.7 | 6.2 | 40.0 | 10.8 | 30.0 | 10.8 | 33.3 | 9.4 | 25.0 | 8.2 |
| | Count Clusters | **42.6** | 2.6 | 27.8 | 0.0 | 20.4 | 9.4 | 20.4 | 2.6 | 11.1 | 4.5 | 9.3 | 6.9 | 3.7 | 2.6 |
| | Count Same Patterns | **25.7** | 7.0 | 15.2 | 3.6 | 10.5 | 2.7 | 6.7 | 2.7 | 9.5 | 1.4 | 3.8 | 2.7 | 1.9 | 1.4 |
| | Find the Different | **29.2** | 7.8 | 16.7 | 3.0 | 20.8 | 3.0 | 14.6 | 3.0 | 2.1 | 3.0 | 0.0 | 0.0 | 6.3 | 0.0 |
| | Find the Same | 21.6 | 5.6 | **23.5** | 8.3 | 9.8 | 5.6 | 9.8 | 2.8 | 7.8 | 2.8 | 2.0 | 2.8 | 3.9 | 5.6 |
| | Reconstruction | **50.0** | 15.4 | 33.3 | 3.4 | 38.1 | 3.4 | 26.2 | 3.4 | 9.5 | 3.4 | 11.9 | 8.9 | 14.3 | 5.8 |
| | Find the Shadow | 14.5 | 7.4 | 15.9 | 5.4 | **17.4** | 3.6 | 7.3 | 2.1 | 7.3 | 2.1 | 10.1 | 2.1 | 7.3 | 5.4 |
| | Pattern and Color Completion | **46.7** | 6.2 | 38.3 | 9.4 | 31.7 | 6.2 | 25.0 | 7.1 | 35.0 | 7.1 | 21.7 | 6.2 | 13.3 | 8.5 |
| | **Sub-Type Overall Results** | **32.9** | 1.0 | 23.3 | 0.9 | 18.6 | 2.3 | 17.4 | 2.9 | 14.1 | 2.7 | 11.3 | 2.3 | 8.8 | 1.5 |
| Visual Tracking | Connect the Lines | 0.0 | 0.0 | 5.3 | 4.3 | 15.8 | 4.3 | 14.0 | 2.5 | 3.5 | 2.5 | 7.0 | 2.5 | **24.6** | 2.5 |
| | Lines Observation | **44.4** | 0.0 | 7.4 | 5.2 | 0.0 | 0.0 | 0.0 | 0.0 | 0.0 | 0.0 | 0.0 | 0.0 | 0.0 | 0.0 |
| | Maze | **26.7** | 2.4 | 11.7 | 6.2 | 23.3 | 2.4 | 21.7 | 2.4 | 8.3 | 2.4 | 21.7 | 2.4 | 23.3 | 4.7 |
| | Metro Map | **27.8** | 7.9 | 13.9 | 3.9 | 2.8 | 3.9 | 0.0 | 0.0 | 13.9 | 10.4 | 2.8 | 3.9 | 5.6 | 3.9 |
| | Recognize Numbers and Letters | 23.2 | 7.4 | 36.2 | 7.4 | 39.1 | 3.6 | 30.4 | 6.2 | 14.5 | 2.1 | 14.5 | 4.1 | 8.7 | 3.6 |
| | **Sub-Type Overall Results** | **21.7** | 3.6 | 16.9 | 4.3 | 20.5 | 2.0 | 16.9 | 1.7 | 8.8 | 2.1 | 22.1 | 1.5 | 14.5 | 2.0 |
| Spatial Perception | 3D Cube Unfold | **33.3** | 6.8 | 19.4 | 7.9 | 8.3 | 6.8 | 8.3 | 6.8 | 8.3 | 6.8 | 11.1 | 10.4 | 2.8 | 3.9 |
| | 3D Pattern Completion | **42.6** | 11.4 | 29.6 | 6.9 | 22.2 | 4.5 | 33.3 | 7.9 | 37.0 | 2.6 | 25.9 | 9.4 | 31.5 | 14.6 |
| | 3D Views | **34.6** | 3.5 | 29.6 | 8.0 | 18.5 | 3.0 | 22.2 | 3.0 | 23.5 | 1.8 | 9.9 | 4.6 | 19.8 | 4.6 |
| | Count 3D Blocks | **19.7** | 5.7 | 10.6 | 2.1 | 6.1 | 2.1 | 6.1 | 4.3 | 10.6 | 4.3 | 3.0 | 2.1 | 6.1 | 4.3 |
| | Paper Folding | 13.9 | 3.9 | 8.3 | 6.8 | 11.1 | 3.9 | 16.7 | 0.0 | 13.9 | 3.9 | 11.1 | 7.9 | **19.4** | 3.9 |
| | **Sub-Type Overall Results** | **29.7** | 0.9 | 20.9 | 2.7 | 13.9 | 2.9 | 18.0 | 1.4 | 19.8 | 0.9 | 11.7 | 1.4 | 16.5 | 2.4 |
| Visual Pattern Recognition | Logic Patterns | **40.5** | 6.7 | 19.1 | 6.7 | 26.2 | 3.4 | 11.9 | 3.4 | 9.5 | 3.4 | 14.3 | 0.0 | 4.8 | 3.4 |
| | Mirroring Patterns | **56.7** | 4.7 | 26.7 | 9.4 | 23.3 | 4.7 | 23.3 | 4.7 | 26.7 | 4.7 | 6.7 | 4.7 | 6.7 | 9.4 |
| | Overlay Patterns | **49.0** | 11.1 | 33.3 | 7.3 | 25.5 | 2.8 | 13.7 | 7.3 | 21.6 | 2.8 | 31.4 | 2.8 | 21.6 | 14.0 |
| | Rotation Patterns | **63.3** | 9.4 | 40.0 | 0.0 | 43.3 | 12.5 | 33.3 | 4.7 | 26.7 | 12.5 | 16.7 | 4.7 | 16.7 | 9.4 |
| | **Sub-Type Overall Results** | **51.0** | 7.3 | 29.4 | 5.8 | 28.8 | 1.9 | 19.0 | 2.5 | 20.3 | 3.3 | 19.0 | 0.9 | 13.1 | 4.9 |
| All | **Overall Results** | **32.1** | 1.6 | 22.2 | 1.0 | 19.2 | 0.7 | 17.6 | 1.8 | 15.1 | 1.3 | 14.7 | 0.8 | 12.4 | 1.7 |

*Table 6.* **Performance (Avg@3) of Open-Source MLLMs on BabyVision.** The best results for each question type are marked in **bold**. Reported values represent the average Pass@1 accuracy across three random runs, accompanied by the standard deviation.

## C.4. RLVR Training Details

We investigate whether Reinforcement Learning with Verifiable Rewards (RLVR) can improve visual reasoning abilities on BabyVision. We use Qwen3-VL-8B-Thinking as the base model and apply RLVR fine-tuning with 1,400 training examples collected following a BabyVision-style pipeline.

**Training Setup.** We train Qwen3-VL-8B-Thinking using GRPO on 8 H800 GPUs for approximately 3 days over 18 epochs. To allow for sufficient exploration, we configure the rollout $n$ to 10, the max response length to 16384, and set the clip range (clip higher) to 0.28. Both the global batch size and rollout batch size are set to 64 within the EasyR1 (Zheng et al., 2025; Sheng et al., 2024) framework. We employ an LLM judge to determine the consistency between the answer extracted from \boxed{} label and the ground truth, using it as the reward signal to guide the model.

Note that the collected training data covers all four major BabyVision task families, yet its difficulty distribution is not completely aligned with BabyVision: the model achieves 34.2% initial accuracy on the RLVR training set, but only 13.1% on BabyVision, when evaluated with the same base model.

**Detailed Results.** The training dynamics are shown in Figure 6, and the per-category results are shown in Figure 5 in the main text. We observe that RLVR is effective on the collected training dataset: both training accuracy and held-out test accuracy consistently improve over the course of training. The model achieves a +4.8-point overall accuracy improvement after RLVR training, with consistent gains across most task subtypes. The sole exception is Visual Tracking, for which RL fine-tuning yields little to even negative improvement. This is likely because visual tracking is the least amenable to verbalization; since RLVR primarily enhances performance by encouraging longer and more structured "thinking-token" reasoning, it provides less benefit on tasks that depend on continuous perceptual tracking rather than language-mediated reasoning. The detailed per-subtype performance comparison is listed in Table 8.

| Type | Sub-Type | Qwen3VL-235B-Thinking | | Qwen3VL-235B-Instruct | | Qwen3VL-32B-Thinking | | Qwen3VL-8B-Thinking | | Qwen3VL-4B-Thinking | |
|---|---|---|---|---|---|---|---|---|---|---|---|
| | | Avg ($\mu$)↑ | Std ($\sigma$)↓ | Avg ($\mu$)↑ | Std ($\sigma$)↓ | Avg ($\mu$)↑ | Std ($\sigma$)↓ | Avg ($\mu$)↑ | Std ($\sigma$)↓ | Avg ($\mu$)↑ | Std ($\sigma$)↓ |
| Fine-grained Discrimination | 2D Pattern Completion | 25.0 | 10.8 | 31.7 | 10.3 | **38.3** | 4.7 | 23.3 | 8.5 | 20.0 | 8.2 |
| | Count Clusters | 27.8 | 0.0 | 16.7 | 4.5 | **35.2** | 5.2 | 20.4 | 2.6 | 16.7 | 4.5 |
| | Count Same Patterns | **15.2** | 3.6 | 11.4 | 0.0 | 11.4 | 2.3 | 2.9 | 2.3 | 3.8 | 1.4 |
| | Find the Different | **16.7** | 3.0 | 4.2 | 3.0 | 6.3 | 5.1 | 0.0 | 0.0 | 0.0 | 0.0 |
| | Find the Same | **23.5** | 8.3 | 11.8 | 4.8 | 7.8 | 2.8 | 3.9 | 2.8 | 11.8 | 4.8 |
| | Reconstruction | **33.3** | 3.4 | 21.4 | 5.8 | 23.8 | 6.7 | 19.1 | 3.4 | 19.1 | 3.4 |
| | Find the Shadow | 15.9 | 5.4 | **24.6** | 2.1 | **24.6** | 5.4 | 11.6 | 2.1 | 8.7 | 6.2 |
| | Pattern and Color Completion | **38.3** | 9.4 | 10.0 | 4.1 | 25.0 | 7.1 | 26.7 | 4.7 | 23.3 | 6.2 |
| | **Sub-Type Overall Results** | **23.3** | 0.9 | 16.4 | 1.3 | 21.1 | 1.8 | 12.7 | 0.6 | 12.1 | 2.3 |
| Visual Tracking | Connect the Lines | 5.3 | 4.3 | 10.5 | 7.4 | 10.5 | 4.3 | **12.3** | 2.5 | 10.5 | 0.0 |
| | Lines Observation | **7.4** | 5.2 | 0.0 | 0.0 | 0.0 | 0.0 | 0.0 | 0.0 | 0.0 | 0.0 |
| | Maze | 11.7 | 6.2 | 11.7 | 4.7 | 5.0 | 4.1 | 20.0 | 0.0 | **28.3** | 6.2 |
| | Metro Map | 13.9 | 3.9 | 13.9 | 3.9 | **19.4** | 3.9 | 5.6 | 3.9 | 5.6 | 7.9 |
| | Recognize Numbers and Letters | **36.2** | 7.4 | 24.6 | 2.1 | 17.4 | 3.6 | 8.7 | 0.0 | 17.4 | 3.6 |
| | **Sub-Type Overall Results** | **16.9** | 4.3 | 15.3 | 0.6 | 8.8 | 2.5 | 10.8 | 1.0 | 14.9 | 1.1 |
| Spatial Perception | 3D Cube Unfold | **19.4** | 7.9 | 13.9 | 3.9 | 16.7 | 6.8 | 5.6 | 3.9 | 8.3 | 6.8 |
| | 3D Pattern Completion | 29.6 | 6.9 | **31.5** | 5.2 | 24.1 | 5.2 | 22.2 | 4.5 | 25.9 | 9.4 |
| | 3D Views | **29.6** | 8.0 | **29.6** | 3.0 | 11.1 | 8.0 | 21.0 | 8.7 | **29.6** | 3.0 |
| | Count 3D Blocks | 10.6 | 2.1 | **21.2** | 9.3 | 7.6 | 2.1 | 7.6 | 5.7 | 12.1 | 4.3 |
| | Paper Folding | 8.3 | 6.8 | 11.1 | 10.4 | 5.6 | 3.9 | **13.9** | 7.9 | 8.3 | 6.8 |
| | **Sub-Type Overall Results** | 20.9 | 2.7 | **23.4** | 2.7 | 12.8 | 1.9 | 15.0 | 3.2 | 19.1 | 1.4 |
| Visual Pattern Recognition | Logic Patterns | 19.1 | 6.7 | **21.4** | 5.8 | **21.4** | 5.8 | 11.9 | 3.4 | 14.3 | 5.8 |
| | Mirroring Patterns | 26.7 | 9.4 | 20.0 | 0.0 | **30.0** | 8.2 | 6.7 | 4.7 | 10.0 | 8.2 |
| | Overlay Patterns | **33.3** | 7.3 | **33.3** | 2.8 | 19.6 | 5.6 | 7.8 | 5.6 | 11.8 | 0.0 |
| | Rotation Patterns | 40.0 | 0.0 | 43.3 | 12.5 | **46.7** | 9.4 | 40.0 | 14.1 | 23.3 | 12.5 |
| | **Sub-Type Overall Results** | **29.4** | 5.8 | **29.4** | 4.2 | 27.5 | 3.2 | 15.0 | 1.9 | 14.4 | 2.5 |
| All | **Overall Results** | **22.2** | 1.0 | 19.5 | 1.4 | 17.4 | 2.0 | 13.1 | 1.1 | 14.6 | 1.2 |

*Table 7.* **Performance (Avg@3) of Qwen3VL Instruct/Thinking on BabyVision.** The best results for each question type are marked in **bold**. Reported values represent the average Pass@1 accuracy across three random runs, accompanied by the standard deviation.

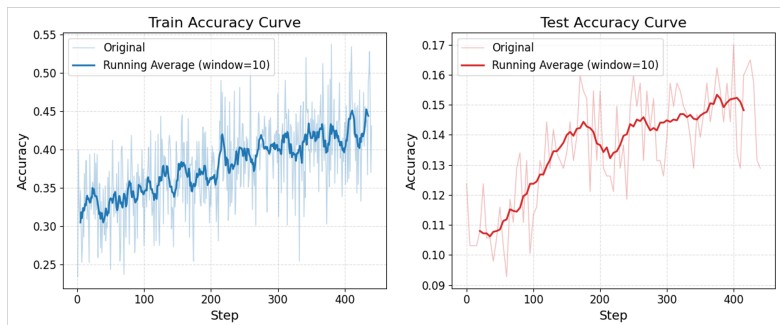

*Figure 6.* GRPO training dynamics for Qwen3-VL-8B-Thinking. Both training accuracy and held-out test accuracy steadily improve during training.

## C.5. CoT Ablation and Attention Probing

To provide causal and mechanistic evidence for the verbalization bottleneck hypothesis, we conduct two experiments on Qwen3-VL-8B-Thinking.

**CoT Ablation.** We force the model to bypass Chain-of-Thought by inserting `<think></think>` tags, keeping the visual encoder, input resolution, and model weights identical. The sole variable is the presence of intermediate linguistic tokens. Results are shown in Table 9.

| Category | w/ CoT | w/o CoT | Δ |
|---|---|---|---|
| Fine-grained Discrim. | 12.7 | **15.1** | +2.4 |
| Visual Tracking | **10.8** | 8.4 | -2.4 |
| Spatial Perception | 15.0 | **18.0** | +3.0 |
| Visual Pattern Recog. | 15.0 | 15.0 | +0.0 |
| Overall | 13.1 | **14.4** | +1.3 |

*Table 9.* CoT ablation on Qwen3-VL-8B-Thinking (accuracy %). Removing CoT selectively improves fine-grained and spatial tasks but hurts Visual Tracking, suggesting CoT helps only when tasks resist verbalization.

| Category | Subtype | Qwen3-VL-8B-Thinking (%) | After GRPO (%) | Δ (%) |
|---|---|---|---|---|
| **Fine-grained Discrimination** | 2D Pattern Completion | 23.3 | **45.0** | +21.7 |
| | Count Clusters | **20.4** | **20.4** | +0.0 |
| | Count Same Patterns | 2.9 | **5.7** | +2.8 |
| | Find the different | 0.0 | **2.1** | +2.1 |
| | Find the same | 3.9 | **15.7** | +11.8 |
| | Reconstruction | 19.1 | **23.8** | +4.7 |
| | Find the shadow | 11.6 | **17.4** | +5.8 |
| | Pattern and Color Completion | 26.7 | **33.3** | +6.6 |
| | **Sub-Type Overall Results** | 12.7 | **19.4** | +6.7 |
| **Visual Tracking** | Connect the lines | **12.3** | 3.5 | -8.8 |
| | Lines Observation | **0.0** | **0.0** | +0.0 |
| | Maze | **20.0** | 15.0 | -5.0 |
| | Metro map | 5.6 | **13.9** | +8.3 |
| | Recognize numbers and letters | 8.7 | **11.6** | +2.9 |
| | **Sub-Type Overall Results** | **10.8** | 9.6 | -1.2 |
| **Spatial Perception** | 3D Cube Unfold | 5.6 | **16.7** | +11.1 |
| | 3D Pattern Completion | 22.2 | **37.0** | +14.8 |
| | 3D Views | 21.0 | **25.9** | +4.9 |
| | Count 3D blocks | 7.6 | **9.1** | +1.5 |
| | Paper Folding | **13.9** | 11.1 | -2.8 |
| | **Sub-Type Overall Results** | 15.0 | **20.9** | +5.9 |
| **Visual Pattern Recognition** | Logic Patterns | 11.9 | **26.2** | +14.3 |
| | Mirroring Patterns | 6.7 | **20.0** | +13.3 |
| | Overlay Patterns | 7.8 | **13.7** | +5.9 |
| | Rotation Patterns | **40.0** | 26.7 | -13.3 |
| | **Sub-Type Overall Results** | 15.0 | **20.9** | +5.9 |
| **Overall Average Accuracy** | | 13.1 | **17.9** | +4.8 |

*Table 8.* GRPO results by subtype (in %). Vertical rules separate category, subtype, and metric groups. For each row, the larger value between the original and GRPO model is bolded.

On several subtypes, disabling CoT produces notable gains: 2D Pattern Completion (+10.0, from 23.3% to 33.3%), Count 3D Blocks (+9.1), Mirroring Patterns (+10.0), and Overlay Patterns (+9.9). If the main bottleneck were in the visual encoder or training data, removing the thinking process would not help in this way. The gains are concentrated on fine-grained and spatial tasks where language is a poor medium for expressing visual details.

**Attention Probing.** We extract the mean attention distribution on the final answer tokens for both settings, as shown in Table 10.

| Setting | CoT Tokens | Prompt Tokens | Image Tokens |
|---|---|---|---|
| w/ CoT | 0.179 | 0.551 | 0.007 |
| w/o CoT | N/A | 0.764 | 0.037 |

*Table 10.* Mean attention weight on final answer tokens by source. With CoT enabled, the model attends almost exclusively to its own generated text, reducing image attention to 0.007; removing CoT increases image attention 5.28×.

Without CoT, vision token attention increases 5.28× (from 0.007 to 0.037). In thinking mode, the model spends most of its attention on its own generated text (0.179) while barely attending to the visual input (0.007). This directly shows how intermediate reasoning in language distracts the model from the original visual features, providing mechanistic support for the verbalization bottleneck claim.

## C.6. Human Age-Group Performance on BabyVision-Mini

Table 11 presents the per-category age-group performance on BabyVision-Mini, a representative 20-question subset of BabyVision. The age-based breakdown does not appear in the main tables because young children (particularly 3-year-olds)

cannot complete the full 388-question benchmark.

| Age Group | Fine-grained Discrim. | Visual Tracking | Spatial Percep. | Pattern Recog. | Overall |
|---|---|---|---|---|---|
| Age-3 | 35 | 40 | 35 | 50 | 40.0 |
| Age-6 | 65 | 60 | 60 | 75 | 65.0 |
| Age-10 | 80 | 85 | 70 | 90 | 81.3 |
| Age-12 | 90 | 85 | 85 | 95 | 88.8 |
| Adult | 95 | 95 | 95 | 100 | 96.3 |
| Gemini3-Pro | 45 | 40 | 50 | 55 | 47.5 |

*Table 11.* Per-category performance (%) on BabyVision-Mini across human age groups and the best MLLM (Gemini3-Pro-Preview). The results show a clear developmental progression. Gemini3-Pro-Preview falls below Age-3 on Visual Tracking and is comparable to Age-3 on Fine-grained Discrimination, despite excelling at knowledge-intensive benchmarks.

The results show a clear developmental progression across all categories. Gemini3-Pro-Preview falls below Age-3 on Visual Tracking and is comparable to Age-3 on Fine-grained Discrimination, despite excelling on knowledge-intensive benchmarks such as MMMU. This confirms that the performance gap is not merely about task difficulty but reflects a fundamental difference in visual processing capabilities between MLLMs and developing human vision.

## D. Detailed Failure Mode Analysis

We identify four systematic failure modes that explain why MLLMs struggle with BabyVision tasks. Representative failure cases from Gemini3-Pro-Preview are illustrated in Figure 7.

**Loss of Fine-Grained Detail.** A pervasive weakness across BabyVision is the degradation of fine-grained visual information, where MLLMs fail to distinguish candidates relying on sub-semantic cues like specific curvature or pixel-level boundary alignment. While humans solve these tasks via direct shape matching—a parallel geometric operation that verifies congruence without intermediate description—MLLMs rely on implicit semantic compression, attempting to discretize continuous shapes into lossy linguistic tokens. This process creates a resolution gap where fine spatial structure is flattened into semantic space, rendering micro-differences indistinguishable and forcing the model to reason through a low-fidelity proxy rather than end-to-end perceptual comparison.

**Loss of Manifold Identity.** We observe a critical failure in topological consistency, where MLLMs struggle to maintain the identity of a continuous manifold (e.g., a winding line) as it interacts with others. Unlike humans, who utilize primitive contour integration to "lock onto" a curve and track it through occlusions, MLLMs attempt to map these 1D manifolds into discrete instruction sequences. Without a persistent, distinct representation of the specific curve, the model faces combinatorial branching at every intersection; consequently, it often "switches tracks" or hallucinates endpoints, revealing an inability to separate overlapping signals or preserve perceptual identity across extended spatial trajectories.

**Failure of Spatial Imagination.** A third bottleneck is the inability to perform mental affine transformations—constructing and manipulating a stable 3D mental model from 2D input (e.g., rotation, projection). While humans solve these tasks via non-verbal mental simulation that retains geometric fidelity, MLLMs attempt to approximate 3D states through descriptive logic: summarizing components and inferring views via language. This fails because language acts as an insufficient coordinate system for volumetric constraints; by substituting physics-compliant rendering with probabilistic text generation, models inevitably hallucinate hidden structures or deduce impossible projections.

**Failure of Visual Pattern Induction.** The fourth challenge is abstract rule acquisition, or the ability to induce generalized transformation rules from sparse examples. Humans readily disentangle abstract relational structures (such as rotation or nesting) from specific visual attributes, but MLLMs frequently conflate style with structure. By approaching induction through attribute enumeration rather than relational mapping, models fixate on spurious surface correlations (e.g., specific colors) instead of the underlying transformation logic, failing to treat objects as variables in a logical operation and missing the compositional generalization required for these tasks.

## E. Case Studies

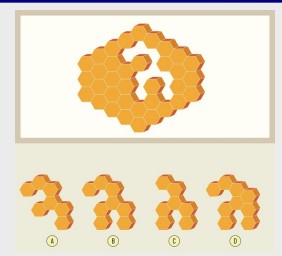

**Question:** Find the missing piece that fits into the empty white space within the hexagonal structure.
**Answer:** B  **Model Answer:** D
**Reasoning:** Bottom island implies split legs; check right then left flank.
Option A has a hexagon sticking out to the right... Option B also has a "shoulder"... Option C has a straight right edge... Option D has an empty right side (a narrow "neck")... Option C is flat on the left... Option B is compact... Option D has a left protrusion...
**Why wrong:** Over-verbalized geometry; misses the exact contour.

(a) Non-verbal fine-grained perception failure

**Question:** Which colored trash bin will the plastic cup, old newspaper, and apple core each go into?
**Answer:** A-blue, B-yellow, C-green  **Model Answer:** A-green, B-yellow, C-blue
**Reasoning:** For plastic cup (A), start at its dot; Follow the line to the left. It curves down and forms the outermost path on the left side of the maze. It continues down, runs along the bottom, and goes up to green (A-green). For apple core (C), the top path goes left, down the second vertical line, and into blue (C-blue). The remaining line links the newspaper (B) to yellow.
**Why wrong:** Track-switching at crossings swaps endpoints.

(b) Failure of manifold identity tracking

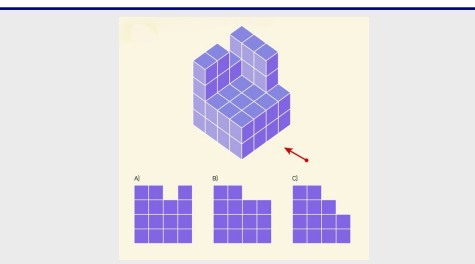

**Question:** Which is the correct view when looking at the 3D structure from the direction indicated by the arrow?
**Answer:** A  **Model Answer:** C
**Reasoning:** Arrow indicates a right-side view; project the silhouette by columns. Columns 1–2 are tallest (tower behind the seat), column 3 is lower (seat), and column 4 is the step. Its height is 2. There is nothing taller behind it. Therefore, the silhouette height is 2. So the profile becomes 4, 4, 3, 2 and the model picks C.
**Why wrong:** Verbal height logic ignores the true 3D structure and occlusion.

(c) Breakdown in 3D spatial imagination

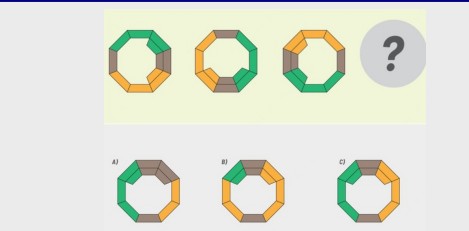

**Question:** Among the three given options, choose the most suitable one to fill in the question mark, so that it shows a certain pattern.
**Answer:** C  **Model Answer:** B
**Reasoning:** Each octagon is composed of 8 equal segments, with two green, two brown, and four orange segments. The brown pair shifts two positions counterclockwise each step, so the next should place brown at the top; B matches that. Option C is rejected because brown appears at the top left and bottom.
**Why wrong:** Focuses on color placement rather than the actual structural transform.

(d) Failure in abstract visual pattern induction

*Figure 7.* Four classic vision-centric challenges for MLLMs. All examples highlight failures caused by compressing perceptual reasoning into language.

**Prompt: Draw a red line to trace the complete line extending from the first animal.**

**Input**    **Ground Truth**    *NanoBanana-Pro*    *Qwen-Image-Edit-2511*

*Sora-2*

**Prompt: The entrance at A is the right one. Please draw the path from A to the exit.**

**Input**    **Ground Truth**    *NanoBanana-Pro*    *Qwen-Image-Edit-2511*

*Veo3*

*Figure 8.* Representative examples of visual reasoning results from different image/video generation models evaluated on BABYVISION-GEN.

