# OpenReview forum: "BabyVision: Visual Reasoning Beyond Language"
_ICML.cc/2026/Conference — ICML 2026 regular_

### Official Review · Reviewer_LYQK · 2026-03-10

**Soundness:** 3
**Presentation:** 3
**Significance:** 3
**Originality:** 3
**Overall Recommendation:** 4
**Confidence:** 4

**Summary:**

This paper introduces BabyVision, a benchmark designed to assess core visual abilities independent of linguistic knowledge of MLLMs, and BabyVision-Gen, an extension for evaluating visual reasoning ability for generative models beyond language output. Empirical results reveal that state-of-the-art models fail on these questions. Moreover, the authors have tried RLVR fine-tuning, which partially alleviates this problem.

**Compliance With Llm Reviewing Policy:**

Affirmed.

**Final Justification:**

After checking the rebuttal, all my concerns have been addressed. Therefore, I will maintain my inital positive score.

**Key Questions For Authors:**

Please check the "weaknesses" section above.

**Limitations:**

No.

**Strengths And Weaknesses:**

**Strengths**
1. This paper is well-written and easy to follow.
2. The motivation is clear and reasonable.
3. The proposed benchmarks are difficult enough and reveal shortcomings of current MLLMs indeed.

**Weaknesses**
1. Detailed analysis across different ages of babies is missing. Actually, we can only see human performance in all tables.
2. Human performance (across different ages) and sora-2 (as the authors have demonstrated some cases generated by sora-2) performance for BabyVision-Gen are missing.
3. A detailed analysis of image models and video models for BabyVision-Gen is also missing.

---

> ### Author Rebuttal · Authors · 2026-03-30
>
> We thank Reviewer `LYQK` for the positive evaluation and for the clear, actionable suggestions.
>
> > W1: Detailed analysis across different ages of babies is missing. We can only see human performance in all tables.
>
> We do provide a developmental trajectory across ages 3, 6, 10, and 12 in Figure 1. The age-based breakdown does not appear in the main tables because young children (particularly 3-year-olds) cannot complete the full 388-question benchmark. We constructed a representative 20-question subset (BabyVision-Mini) for younger groups. Below are the per-category results:
>
> | Age Group/BabyVision-Mini | Fine-grained Discrim. | Visual Tracking | Spatial Percep. | Pattern Recog. | Overall |
> | --- | --- | --- | --- | --- | --- |
> | Age-3 | 35 | 40 | 35 | 50 | 40.0 |
> | Age-6 | 65 | 60 | 60 | 75 | 65.0 |
> | Age-10 | 80 | 85 | 70 | 90 | 81.3 |
> | Age-12 | 90 | 85 | 85 | 95 | 88.8 |
> | Adult | 95 | 95 | 95 | 100 | 96.3 |
> | Gemini3-Pro  | 45 | 40 | 50 | 55 | 47.5 |
>
> The results show a clear developmental progression. Gemini3-Pro-Preview falls below Age-3 on Visual Tracking and is comparable to Age-3 on Fine-grained Discrimination. We will include this table in the revision.
>
> > W2: Human performance and Sora-2 performance for BabyVision-Gen are missing.
>
> We did evaluate video generation models including Sora-2 and Veo3. Qualitatively, these models do attempt to solve the tasks through visual reasoning in video space (e.g., tracing a path frame by frame through a maze), which is an encouraging sign. However, current video models suffer from two problems that prevent reliable quantitative evaluation: (1) image consistency degrades across frames, with the original image structure often distorted in later frames, and (2) video duration limitations prevent models from completing complex tasks. **The two problem makes it difficult to fairly score a video generation model.** As a result, we discuss video generation as a promising future direction in Section 6.3 rather than reporting quantitative results. See Figure 8 (Appendix) for qualitative examples.
>
> As for the Human baselines, we had 10 adult testers provide hand-drawn solutions on printed BabyVision-Gen questions. Human accuracy is 98.1%, confirming a large gap with all generation models (best: NanoBanana-Pro at 18.3%). We will include this in the revision.
>
> > W3: A detailed analysis of image models and video models for BabyVision-Gen is also missing.
>
> For image models, the comparison is in Table 2: NanoBanana-Pro leads (overall 18.3%), GPT-Image-1.5 is weaker but balanced (9.8%), and Qwen-Image-Edit struggles across the board (4.8%). Image models can preserve the original image and add annotations on top, which makes evaluation straightforward.
>
> For video models (Sora-2, Veo3, Wan-2.2), the situation is different. As discussed in W2, these models do attempt visual reasoning through video (e.g., progressively tracing a path), but they suffer from image consistency degradation in later frames and video duration limits that prevent completing complex tasks. The generated videos often show structural changes to the original image, making quantitative evaluation unreliable. We therefore present video generation as a discussion direction (Section 6.3) with qualitative examples (Figure 8) rather than quantitative results. As video generation models improve in temporal consistency and controllability, we expect them to become a viable evaluation modality for BabyVision-Gen.

---

> > ### Author Rebuttal · Reviewer_LYQK · 2026-04-04
> >
> > All my concerns have been addressed, and thus I will maintain my positive score. The authors are expected to revise the paper based on this discussion.

---

### Official Review · Reviewer_FQqR · 2026-03-11

**Soundness:** 3
**Presentation:** 4
**Significance:** 2
**Originality:** 1
**Overall Recommendation:** 2
**Confidence:** 5

**Summary:**

BabyVision introduces a benchmark of 388 manually curated questions across 22 subtypes in four domains (fine-grained discrimination, visual tracking, spatial perception, visual pattern recognition), targeting perceptual abilities that are pre-linguistic in humans but apparently absent in current MLLMs. The paper benchmarks 11 frontier models and compares them against human participants. It additionally proposes BabyVision-Gen, a generative variant where models must annotate images with visual answers (traces, circles, etc.) rather than produce text. A small RLVR experiment using GRPO on Qwen3-VL-8B provides preliminary evidence that RL fine-tuning helps on most subtypes but fails on visual tracking, which the authors attribute to the "verbalization bottleneck."

**Compliance With Llm Reviewing Policy:**

Affirmed.

**Final Justification:**

The rebuttal has partially addressed the concern about the goal of the dataset, although I am still not sure of how useful it is to compare VLMs to different states of human perception development. However, my concerns regarding dataset size, overlap with prior work, and leakage/saturation concerns were not addressed. I am keeping my initial score (Reject/2).

**Key Questions For Authors:**

No questions.

**Limitations:**

Limited dataset size and novelty.

**Strengths And Weaknesses:**

**Strengths**
1. Dataset with novel task coverage. While some overlap with prior benchmarks exists (see weaknesses), several subtypes here are likely absent from existing VLM evaluation suites.
2. Requiring models to express solutions through image annotation rather than text output is an underexplored evaluation modality.

3. The GRPO experiment on Qwen3-VL-8B is a useful proof of concept. The finding that RL fine-tuning yields consistent gains on most subtypes (+4.8 overall) but flat or negative gains on visual tracking is a substantive and non-obvious result.

**Weaknesses**
1. Limited novelty. The paper positions BabyVision as filling a gap, but prior work covers substantially overlapping ground. IQBench and VRIQ have both tested VLMs on IQ-style visual reasoning tasks that are the same/highly similar to BabyVision. Hsu et al. specifically studied VLM maze-solving ability, one of BabyVision's subtypes.

2. A substantial fraction of the tasks are structurally amenable to procedural generation at scale. With only 388 total questions and some subtypes having as few as 9-12 examples, the benchmark is too small to rule out saturation. Releasing a procedural generator would have been a more durable contribution aligned with the paper's stated goals.

3. Section 6.1 failure mode analysis is unsupported. The four failure modes appear to be qualitative interpretations rather than conclusive evidence. No ablation, probing experiment, or controlled manipulation isolates whether these are actual failure causes versus post-hoc narratives. The claim that fine-grained discrimination fails due to "lossy linguistic token compression" rather than visual encoder resolution or training distribution is not tested.

---

> ### Author Rebuttal · Authors · 2026-03-30
>
> We thank Reviewer FQqR for the detailed review and for recognizing BabyVision’s novel task coverage, its underexplored generative evaluation setting, and the RLVR result on visual tracking.
>
> > W1: Limited novelty. IQBench and VRIQ ...
>
> We respectfully disagree. Two cited references do not support this claim.
>
> First, VRIQ (arXiv:2602.05382) was posted on Feb 5, 2026, eight days after the ICML 2026 submission deadline (Jan 28, 2026). It did not exist when we submitted, so it cannot be used to assess novelty at submission time. We are happy to discuss it in the camera-ready. Second, “Hsu et al.” is mentioned without a citation.
>
> This leaves IQBench (arXiv:2505.12000, May 2025) as the only identifiable prior work. IQBench and BabyVision target fundamentally different constructs. IQBench is an adult IQ benchmark centered on symbolic logic and rule induction. BabyVision evaluates pre-linguistic perceptual primitives acquired by children ages 3–12, grounded in developmental psychology. This is a difference in kind, not degree.
>
> IQBench contains 10 categories, of which 6 (300/500 questions) are language or number based and irrelevant to BabyVision. Among its 4 visual categories, the apparent overlap is only superficial. IQBench’s 3D spatial tasks require abstract mental rotation and deductive reasoning over complex geometry. BabyVision’s spatial tasks test concrete perception: counting occluded blocks, matching cubes to unfolded nets, and predicting folded-paper outcomes—tasks that 10-year-old children solve reliably (86.2% in our study). Similarly, IQBench Figure Series measures symbolic rule induction, whereas BabyVision’s visual pattern tasks test basic perceptual transformations such as rotation, mirroring, and overlay.
>
> The only genuine format-level overlap is Logic Patterns: 14 questions, or 3.6% of BabyVision. The remaining **96.4%** measures different abilities with different formats. Beyond task coverage, BabyVision also contributes: (1) a developmental-psychology-based taxonomy, (2) human baselines across ages 3–12 and adults, (3) BabyVision-Gen for generative evaluation, and (4) systematic failure analysis plus RLVR study.
>
> > W2: Many tasks are procedurally generable; 388 questions is too small; a generator would be more durable.
>
> All subtypes rely on a wide variety of real-world images from the internet (e.g., Find the Shadow, Find the Same, Reconstruction), which are difficult to synthesize while preserving realistic visual properties. BabyVision intentionally includes such naturalistic tasks. Even within the same subtype, images often come from diverse sources, making a unified procedural generator difficult to build.
>
> We also do not believe saturation is imminent. Many subtypes remain low scores (e.g., Find the Same 26.5%, Count 3D Blocks 20.5%, Lines Observation 0% for most models). More importantly, BabyVision spans 22 subtypes across four categories, covering largely independent skills such as fine-grained comparison, 3D reasoning, and line tracking. Saturating one or two subtypes would not imply broad mastery.
>
> > W3: Section 6.1 failure analysis is unsupported; the “lossy linguistic token compression” claim is untested.
>
> We added both a controlled manipulation and a mechanistic probe to directly test the claim.
>
> **CoT ablation.** We forced Qwen3-VL-8B-Thinking to bypass Chain-of-Thought by inserting `<think></think>`, keeping encoder, resolution, and weights unchanged. The only change is removing intermediate linguistic reasoning. Non-thinking performs better overall (14.4% vs. 13.1%), with gains concentrated on fine-grained and spatial tasks where language is a poor medium for visual detail: 2D Pattern Completion +10.0, Count 3D Blocks +9.1, Mirroring +10.0, Overlay +9.9. If the main bottleneck were the visual encoder or training data, disabling CoT should not help in this way. This provides causal evidence that language-mediated reasoning can discard fine visual information.
>
> **Attention probing.** We measured mean attention on final answer tokens. In thinking mode, attention to self-generated text / prompt / vision tokens is 0.179 / 0.551 / 0.007. In non-thinking mode, it is N/A / 0.764 / 0.037. Thus, attention to vision tokens increases 5.28× when CoT is removed. In thinking mode, the model attends substantially to its own generated text while barely attending to the image. This directly supports the proposed mechanism.
>
> Further, RLVR improves most categories but not Visual Tracking (Section 6.2). Since RLVR mainly strengthens language-mediated reasoning, its failure on tracking tasks independently supports the verbalization bottleneck. On these near-zero tracking tasks, generation models sometimes produce correct visual traces (Appendix Figure 8), providing additional evidence.
>
> We have revised Section 6.1 to include the experiment results.

---

> > ### Author Rebuttal · Reviewer_FQqR · 2026-04-01
> >
> > W3 has been fully addressed.
> >
> > I have concerns regarding W1 and W2.
> >
> > W1. Novelty and overlap with existing benchmarks
> >
> > I think think there are multiple works that cover individual tasks in BabyVision. Here's what I have from the top of my head:
> >
> > 1. Maze: Hsu et al. What Makes A Maze Look Like A Maze? 2025
> > 2. 3D Views: Pham and Nguyen et al. IQBench: How “Smart” Are Vision-Language Models? A Study with Human IQ Tests. Figure 1(h), 2025
> > 3. Counting: Guo et al. Your Vision-Language Model Can’t Even Count to 20: Exposing the Failures of VLMs in Compositional Counting 2025
> >
> >
> > W2. Dataset size and generator
> > Since the dataset collected from the internet, they could be part of the VLMs' training data as well. A generator should be possible for most of the sub types of BabyVision. Naturalistic variations are important but as the data is collected from the internet, future VLMs could overfit inadvertently due to inclusion in pretraining data.

---

> > > ### Author Response · Authors · 2026-04-03
> > >
> > > ### Response to Reviewer FQqR's Follow-up
> > >
> > > We thank the reviewer for the continued discussion and for providing the specific references.
> > >
> > > **W1 follow-up: Overlap with Hsu et al., IQBench, and Guo et al.**
> > >
> > > We have carefully read all three cited works. Briefly: Hsu et al. studies visual abstraction and schema grounding, where "maze" is one of 12 abstract concepts and the task is recognizing abstract representations (e.g., a maze made of coffee beans), not solving or navigating mazes. IQBench's 3D SPRT tests abstract deductive spatial reasoning, not the concrete perceptual abilities BabyVision targets. Guo et al. tests compositional counting of simple 2D shapes (circles, triangles, squares with no overlap on a blank canvas), which is different from BabyVision's counting tasks that require visual discrimination, grouping, and 3D understanding under occlusion.
> > >
> > > We want to step back and clarify what BabyVision actually aims to do. The core contribution is not any individual task. **It is the identification and systematic evaluation of a previously overlooked problem: current MLLMs lack the pre-linguistic visual primitives that human children develop before age 12. **BabyVision is the first benchmark to ground this evaluation in developmental psychology, organize 22 subtypes into a principled taxonomy of early-vision abilities, provide human baselines across age groups (3 to 12 and adults), and reveal a 44.4% gap between the best model and humans. We further propose BabyVision-Gen as a new evaluation paradigm and provide causal evidence for the verbalization bottleneck through CoT ablation and attention probing.
> > >
> > > The fact that individual visual skills (maze, counting, 3D reasoning) have been studied in isolation by scattered prior works does not diminish the value of a unified evaluation framework. By the same logic, MMMU would lack novelty because individual subject exams already exist, or BLINK would lack novelty because individual perception tasks have been studied. The novelty lies in the framework, the developmental grounding, and the findings, not in any single task type.
> > >
> > > **W2 follow-up: Data contamination concern.**
> > >
> > > We agree that procedural generation is a good idea for long-term benchmark durability, and we are developing a generator to enhance our benchmark.
> > >
> > > However, we respectfully note that the data contamination concern is not well-founded for BabyVision. **All 388 questions and answers are entirely human-crafted through our curation pipeline (Section 3.1).** Even if some source images happen to appear in pretraining data, the specific questions, answer options, and reasoning required are unique to our benchmark and would not be memorized. More importantly, if data leakage were actually occurring, we would expect models to perform well on BabyVision. The opposite is true: the best model scores only 49.7%, and most models fall below 3-year-old children. This performance profile is strong evidence against contamination. Data contamination is a general concern for any benchmark that uses images from the internet, and it is not specific to BabyVision. Holding BabyVision to a standard that is not applied to other benchmarks would be unfair.

---

### Official Review · Reviewer_QM4D · 2026-03-11

**Soundness:** 3
**Presentation:** 3
**Significance:** 3
**Originality:** 3
**Overall Recommendation:** 4
**Confidence:** 3

**Summary:**

This paper introduces BABYVISION, a benchmark designed to evaluate the core visual reasoning abilities of MLLMs independent of linguistic knowledge. The benchmark contains 388 questions across 22 subclasses covering skills such as visual discrimination, tracking, spatial perception, and pattern recognition. Experiments on several state-of-the-art MLLMs reveal a substantial gap between model performance and human baselines, suggesting that current models still struggle with fundamental visual reasoning abilities despite strong performance on knowledge-intensive multimodal tasks.

**Compliance With Llm Reviewing Policy:**

Affirmed.

**Final Justification:**

Most of my concerns have been addressed during the rebuttal. I keep the positive rating.

**Key Questions For Authors:**

- Why are these early-vision abilities important for MLLMs?

**Limitations:**

While the paper introduces BABYVISION to evaluate fundamental visual reasoning abilities of MLLMs, the benchmark is relatively small (388 questions), which may limit the statistical robustness and coverage of the evaluation. In addition, although the work reveals significant performance gaps between models and humans, it mainly provides diagnostic analysis and only briefly explores improvement through methods such as RLVR.

**Strengths And Weaknesses:**

### Strengths
- The paper highlights an underexplored issue: the discrepancy between strong performance on knowledge-intensive multimodal tasks and weak performance on basic visual perception abilities.
- The experiments show a large performance gap between humans and current MLLMs, and the analysis highlights specific failure patterns such as identity tracking failures and spatial reasoning errors.
- The authors further explore a potential mitigation direction by applying RLVR to improve MLLMs’ performance on the BABYVISION benchmark, providing a useful reference for future research on addressing the limitations revealed by the benchmark.

### Weaknesses
- While the paper demonstrates that current models perform poorly on tasks designed to mimic early human visual cognition, it is unclear whether these abilities are necessary or beneficial for solving real-world multimodal tasks.
- The dataset contains only 388 questions, which is quite small for evaluating modern large models. This raises concerns about statistical robustness and whether the results generalize to broader visual reasoning capabilities.
- Although the tasks are motivated by developmental psychology, the paper provides limited discussion on how well the benchmark reliably measures the intended visual abilities or whether certain tasks may still allow shortcuts.

---

> ### Author Rebuttal · Authors · 2026-03-30
>
> We thank Reviewer QM4D for the constructive review and for recognizing the underexplored issue that BabyVision addresses.
>
> > W1: It is unclear whether these early-vision abilities are necessary or beneficial for solving real-world multimodal tasks.
>
> Q1: Why are these early-vision abilities important for MLLMs?
> We argue that early-vision abilities matter for three reasons.
> First, safety-critical applications. MLLMs are being deployed in autonomous driving, medical imaging, and robotic manipulation, all of which demand reliable spatial perception, object tracking, and fine-grained discrimination. A model that cannot track lines through intersections or count 3D blocks under occlusion poses real deployment risks.
> Second, foundation for higher-level reasoning. Developmental psychology shows that early-vision abilities are prerequisites for later cognitive functions(Spelke, E. S. (2000). Core knowledge. American Psychologist, 55(11), 1233–1243. ). We observe similar patterns in our experiments: models that fail at basic tracking also make systematic errors on complex visual reasoning in benchmarks, where graph and diagram understanding depends on the same spatial and tracking abilities.
> Third, benchmark validity. MMStar showed that models can score 42.9% on MMMU without visual input. BabyVision provides a complementary diagnostic that isolates the visual component, helping the community determine whether gains on existing benchmarks reflect genuine visual understanding or linguistic shortcuts.
>
> > W2: The dataset contains only 388 questions, which is quite small for evaluating modern large models.
>
> The small size is a deliberate consequence of our rigorous curation pipeline (Section 3.1). Every question must pass manual annotation, double-blind expert review, and a consensus requirement. Relaxing quality control to increase volume would undermine the diagnostic purpose. For context, several widely-adopted diagnostic benchmarks follow a similar philosophy: BLINK has subtypes with as few as 50 samples, MMVP uses only 300 image pairs, Winoground has 400 examples, and IQBench has 500 samples.
> Regarding robustness, the 44.4% absolute gap between the best model (49.7%) and human performance (94.1%) is too large to be a statistical artifact. The consistently low standard deviations across three evaluation runs (Tables 1 and 3) further confirm that model rankings and subtype-level conclusions are stable despite the smaller sample size.
>
> > W3: Limited discussion on whether the benchmark reliably measures intended abilities or whether tasks may allow shortcuts.
>
> We took several explicit steps to prevent shortcut exploitation during benchmark design.
> 65.2% of questions are fill-in-the-blank rather than multiple choice, reducing random guessing effectiveness (Section 3.2). For multiple-choice questions, we verified balanced answer distributions. Every question was verified by two independent experts to ensure answers are derived from visual analysis rather than background knowledge (Section 3.1). The small standard deviations in Table 1 (typically 0 to 5%) indicate stable rather than noisy model performance.
> We further tested two MLLMs (GPT-5.2, Gemini3-Pro-Preview) on BabyVision questions with text only (no images). Average accuracy drops to 10.3%, confirming that questions cannot be solved through linguistic priors alone. We will include these results in the revision.

---

> > ### Author Rebuttal · Reviewer_QM4D · 2026-04-02
> >
> > Most of my concerns have been addressed, and I currently have no other questions. I keep the rating.

---

### Official Review · Reviewer_4XGd · 2026-03-16

**Soundness:** 3
**Presentation:** 3
**Significance:** 3
**Originality:** 3
**Overall Recommendation:** 4
**Confidence:** 3

**Summary:**

This paper introduces BABYVISION, a benchmark for evaluating pre-linguistic visual reasoning skills that current MLLMs often fail due to overreliance on linguistic priors and semantic shortcuts. Evaluating 13 frontier models against human age-group baselines, including a pixel-based generation setting, the study finds a large performance gap, with most models performing below the level of a 6-year-old.

**Compliance With Llm Reviewing Policy:**

Affirmed.

**Key Questions For Authors:**

see weakness

**Limitations:**

see weakness

**Strengths And Weaknesses:**

Strenghts:
1. Important and timely motivation. The paper highlights a meaningful gap in current MLLM evaluation: models that perform well on language-heavy, knowledge-intensive benchmarks may still lack basic visual reasoning skills.
2. Clear benchmark contribution. BABYVISION is well-motivated and organized around a coherent set of perceptual primitives, making it a useful diagnostic benchmark rather than just another general-purpose test set.
3. Compelling empirical findings. The reported gap between humans and state-of-the-art MLLMs is large and consistent enough to support the paper’s core claim that foundational visual competence remains a major weakness.
4. Interesting methodological extension. BABYVISION-GEN is a thoughtful addition that explores whether some visual reasoning tasks are better evaluated through direct image-based generation rather than text answers.
5. Useful failure analysis. The paper goes beyond reporting aggregate scores and attempts to categorize failure modes, which increases its value for future research.

Weakness:
1. Limited benchmark scale. Although the dataset appears carefully curated, the overall size of 388 questions is still relatively small, which may limit the strength of broad conclusions and the stability of subtype-level analysis.
2. Core explanation is suggestive rather than fully established. The “verbalization bottleneck” hypothesis is plausible, but the current evidence is not sufficient to conclusively isolate it from other possible causes such as visual encoder limitations or training data mismatch.
3. BABYVISION-GEN is still somewhat preliminary. While conceptually promising, current model performance is very low, making it harder to disentangle failures of reasoning from failures of precise visual generation or execution.
4. Limited intervention evidence. The RLVR experiment is interesting but modest in effect, and it does not yet provide a strong path toward solving the underlying perceptual limitations identified by the benchmark.s

---

> ### Author Rebuttal · Authors · 2026-03-30
>
> We thank Reviewer 4XGd for the thoughtful evaluation and for recognizing BabyVision's important motivation, clear benchmark design, compelling empirical findings, and useful failure analysis.
>
> > W1: Limited benchmark scale. The overall size of 388 questions is relatively small, which may limit broad conclusions and subtype-level stability.
>
> The small size is a deliberate consequence of our rigorous curation pipeline (Section 3.1). Every question must pass manual annotation, double-blind expert review, and a consensus requirement. Relaxing quality control to increase volume would undermine the diagnostic purpose. For context, several widely-adopted diagnostic benchmarks follow a similar philosophy: BLINK has subtypes with as few as 50 samples, MMVP uses only 300 image pairs, Winoground has 400 examples, and IQBench has 500 samples.
> Regarding robustness, the 44.4% absolute gap between the best model (49.7%) and human performance (94.1%) is too large to be a statistical artifact. The consistently low standard deviations across three evaluation runs (Tables 1 and 3) further confirm that model rankings and subtype-level conclusions are stable despite the smaller sample size.
>
> > W2: The "verbalization bottleneck" hypothesis is plausible but not sufficient to isolate it from visual encoder limitations or training data mismatch.
>
> We agree that the verbalization bottleneck is one of several contributing factors, and we do not claim it as the sole explanation. To better separate it from alternatives, we conducted two experiments during the rebuttal period:
>
> CoT ablation. We ablated Qwen3-VL-8B-Thinking by injecting a `<think></think>` tag to force the model to skip Chain-of-Thought and directly output the answer, keeping the visual encoder, input resolution, and model weights identical. The sole variable is the presence of intermediate linguistic tokens.
>
> | Category | Thinking (%) | Non-thinking (%) | Delta |
> |---|---|---|---|
> | Fine-grained Discrimination | 12.7 | 15.1 | +2.4 |
> | Visual Tracking | 10.8 | 8.4 | -2.4 |
> | Spatial Perception | 15.0 | 18.0 | +3.0 |
> | Visual Pattern Recognition | 15.0 | 15.0 | 0.0 |
> | Overall | 13.1 | 14.4 | +1.3 |
>
> On several subtypes, disabling CoT produces notable gains: 2D Pattern Completion (+10.0, from 23.3% to 33.3%), Count 3D Blocks (+9.1), Mirroring Patterns (+10.0), Overlay Patterns (+9.9). If the bottleneck were in the visual encoder or training data, removing the thinking process would not help. The gains are concentrated on fine-grained and spatial tasks where language is a poor medium for expressing visual details.
>
> Attention probing. We extracted the mean attention distribution on the final answer tokens for both settings.
>
> | Setting | Self-generated text | Prompt text | Vision tokens |
> |---|---|---|---|
> | Thinking | 0.179 | 0.551 | 0.007 |
> | Non-thinking | N/A | 0.764 | 0.037 |
>
> Without CoT, vision token attention increases 5.28x (from 0.007 to 0.037). In thinking mode, the model spends most of its attention on its own generated text (0.179) while barely looking at the visual input (0.007). This directly shows how intermediate reasoning in language distracts the model from the original visual features.
>
> >  W3: BabyVision-Gen is still preliminary. Current model performance is very low, making it hard to disentangle reasoning failures from generation failures.
>
> We agree that BabyVision-Gen is at an early stage, and we position it as an exploratory contribution rather than a fully mature evaluation framework. Its primary goal is to demonstrate that visual generation is a viable alternative evaluation modality.
> That said, we can partially disentangle the two types of failures. On tasks like Find the Different and Count Same Patterns, NanoBanana-Pro achieves 35.4% and 31.2%, which are comparable to or higher than some MLLMs on the same tasks via text output. This suggests the generation model is performing visual reasoning, not just failing at image generation. Conversely, on Connect the Lines and Maze, all generation models score 0%, which likely reflects both reasoning and generation difficulties combined. We have also added human baselines for BabyVision-Gen (see LYQK-W2).
>
> >  W4: The RLVR experiment is modest in effect and does not provide a strong path toward solving perceptual limitations.
>
> We designed the RLVR experiment as a diagnostic probe, not a proposed solution. The main value is the asymmetric pattern: RLVR helps most categories but fails on Visual Tracking. Since RLVR works by encouraging more structured language reasoning, this independently validates the verbalization bottleneck hypothesis. We discuss visual externalization (Section 6.3) as a more promising long-term direction and will clarify this framing in the revision.

---

> > ### Author Rebuttal · Reviewer_4XGd · 2026-04-01
> >
> > N/A

---

### Decision · Program_Chairs · 2026-04-30

**Decision:**

Accept (regular)

**Comment:**

This paper introduces BabyVision and BabyVision-Gen to investigate a fundamental limitation of current multimodal large language models, namely the verbalization bottleneck that constrains visual reasoning. The motivation is clear and compelling, and the proposed benchmark highlights a substantial gap between human and model performance on tasks that are difficult to solve through language. The generative evaluation protocol is novel and reduces reliance on textual space, offering a more direct probe of visual understanding. In addition, the analysis of the language bottleneck, supported by CoT ablations and attention shifts toward visual tokens, provides useful empirical insight into how current models process visual information.

At the same time, the paper has several notable limitations. The dataset is relatively small, which raises concerns about statistical reliability and generalizability. Potential data contamination is not convincingly ruled out, and the current argumentation is insufficient to ensure evaluation integrity. Furthermore, the work primarily focuses on diagnosis and analysis, with limited technical contribution in terms of concrete methods or solutions. Overall, I assign a weak accept (low priority: accept if there is room in the program), as the paper offers valuable insights and a thought-provoking benchmark, but would benefit from stronger experimental rigor and clearer positioning of its technical contributions.